# Nutrient levels and trade-offs control diversity in a serial dilution ecosystem

Amir Erez[1†], Jaime G Lopez[2†], Benjamin G Weiner[3], Yigal Meir[4], Ned S Wingreen[1,2]*

[1]Department of Molecular Biology, Princeton University, Princeton, United States; [2]Lewis-Sigler Institute for Integrative Genomics, Princeton University, Princeton, United States; [3]Department of Physics, Princeton University, Princeton, United States; [4]Department of Physics, Ben Gurion University of the Negev, Beersheba, Israel

**Abstract** Microbial communities feature an immense diversity of species and this diversity is linked to outcomes ranging from ecosystem stability to medical prognoses. Yet the mechanisms underlying microbial diversity are under debate. While simple resource-competition models don't allow for coexistence of a large number of species, it was recently shown that metabolic trade-offs can allow unlimited diversity. Does this diversity persist with more realistic, intermittent nutrient supply? Here, we demonstrate theoretically that in serial dilution culture, metabolic trade-offs allow for high diversity. When a small amount of nutrient is supplied to each batch, the serial dilution dynamics mimic a chemostat-like steady state. If more nutrient is supplied, community diversity shifts due to an 'early-bird' effect. The interplay of this effect with different environmental factors and diversity-supporting mechanisms leads to a variety of relationships between nutrient supply and diversity, suggesting that real ecosystems may not obey a universal nutrient-diversity relationship.

*For correspondence:
wingreen@princeton.edu

†These authors contributed equally to this work

Competing interests: The authors declare that no competing interests exist.

## Introduction

Microbial communities feature an immense diversity of organisms, with the typical human gut microbiota and a liter of seawater both containing hundreds of distinct microbial types (*Lloyd-Price et al., 2016*; *Ladau et al., 2013*; *Weigel and Pfister, 2019*). These observations appear to clash with a prediction of some resource-competition models, known as the competitive-exclusion principle – namely, that steady-state coexistence is possible for only as many species as resources (*Levin, 1970*; *Armstrong and McGehee, 1980*). This conundrum is familiarly known as the 'paradox of the plankton' (*Hutchinson, 1961*). Solving this paradox may provide one key to predicting and controlling outcomes ranging from ecosystem stability to successful cancer treatments in humans (*Ptacnik et al., 2008*; *van Elsas et al., 2012*; *Taur et al., 2014*; *Stein et al., 2013*). *Chesson, 2000* classified mechanisms that purport to solve this paradox into two broad categories: *stabilizing* and *equalizing*. Stabilizing mechanisms prevent extinction by allowing species to recover from low populations, whereas equalizing mechanisms slow extinction by minimizing fitness differences between species.

Many possible solutions of the paradox that rely on stabilizing mechanisms have been offered: (i) interactions between microbes, such as cross-feeding or antibiotic production and degradation (*Goyal and Maslov, 2018*; *Kelsic et al., 2015*), (ii) spatial heterogeneity (*Murrell and Law, 2003*; *Tilman, 1994*), (iii) persistent non-steady-state dynamics (*Hutchinson, 1961*), and (iv) predation (*Thingstad, 2000*). Equalizing mechanisms have been studied through neutral theory, in which all species are assumed to have equal fitness (*Hubbell, 2005*), and recent work has proposed resource-competition models that self-organize to a neutral state (*Posfai et al., 2017*). Many proposed

**eLife digest** In most environments, organisms compete for limited resources. The number and relative abundance of species that an ecosystem can host is referred to as 'species diversity'. The competitive-exclusion principle is a hypothesis which proposes that, in an ecosystem, competition for resources results in decreased diversity: only species best equipped to consume the available resources thrive, while their less successful competitors die off. However, many natural ecosystems foster a wide array of species despite offering relatively few resources.

Researchers have proposed many competing theories to explain how this paradox can emerge, but they have mainly focused on ecosystems where nutrients are steadily supplied. By contrast, less is known about the way species diversity is maintained when nutrients are only intermittently available, for example in ecosystems that have seasons.

To address this question, Erez, Lopez et al. modeled communities of bacteria in which nutrients were repeatedly added and then used up. Depending on conditions, a variety of relationships between the amount of nutrient supplied and community diversity could emerge, suggesting that ecosystems do not follow a simple, universal rule that dictates species diversity. In particular, the resulting communities displayed a higher diversity of microbes than the limit imposed by the competitive-exclusion principle.

Further observations allowed Erez, Lopez et al. to suggest guiding principles for when diversity in ecosystems will be maintained or lost. In this framework, 'early-bird' species, which rapidly use a subset of the available nutrients, grow to dominate the ecosystem. Even though 'late-bird' species are more effective at consuming the remaining resources, they cannot compete with the increased sheer numbers of the 'early-birds', leading to a 'rich-get-richer' phenomenon.

Oceanic plankton, arctic permafrost and many other threatened, resource-poor ecosystems across the world can dramatically influence our daily lives. Closer to home, shifts in the microbe communities that live on the surface of the human body and in the digestive system are linked to poor health. Understanding how species diversity emerges and changes will help to protect our external and internal environments.

solutions for the paradox assume a chemostat framework wherein nutrients are continuously supplied and there is a continuous removal of biomass and unused nutrients (*Palmer, 1994*). However, in nature nutrients are rarely supplied in a constant and continuous fashion. In particular, seasonal variation is ubiquitous in ecology, influencing systems ranging from oceanic phytoplankton communities (*Chang, 2003*) to the microbiota of some human populations (*Smits et al., 2017*). How does a variable nutrient supply influence diversity?

Existing literature on seasonality focuses on stabilizing mechanisms and generally finds that seasonality either promotes or has little effect on coexistence (*Chesson, 1994*). But do these conclusions extend to equalizing mechanisms? To address this question, we consider a known resource-competition model that permits high diversity at steady state due to the equalizing effects of metabolic trade-offs, which assume that microbes have a limited enzyme production capacity they must apportion. Here, we investigate the equalizing effect of metabolic trade-offs in the context of serial dilution, to reflect a more realistic variable nutrient supply.

Serial dilution, in which cultures of bacteria are periodically diluted and supplied with fresh nutrients, is well-established as an experimental approach. For example, the bacterial populations in the Lenski long-term evolution experiment (*Lenski and Travisano, 1994*), experiments on community assembly (*Goldford et al., 2018*), and antibiotic cross-protection (*Yurtsev et al., 2016*) were all performed in serial dilution. While previous models of serial dilution have characterized competition between small numbers of species with trade-offs in their growth characteristics (*Stewart and Levin, 1973*; *Smith, 2011*), the theoretical understanding of diversity in serial dilution is much less developed than for chemostat-based steady-state growth.

Here, we show that under serial dilution, metabolic trade-offs can still support high diversity communities, but that this coexistence is now sensitive to environmental conditions. Interestingly, seasonality can both increase and decrease diversity in our model, contrasting what has been observed for stabilizing mechanisms. In particular, we find a surprising dependence between community

diversity and the amount of nutrient provided to the community. These changes in diversity are driven by an 'early-bird' effect in which species that efficiently consume nutrients that are initially more abundant gain a population advantage early in the batch. To our knowledge, this is the first time this effect has been identified as a major influencer of diversity in seasonal ecosystems.

This dependence between community diversity and the supplied nutrient concentration allowed us to explore an unresolved question in ecology (*Tilman, 1982*; *Abrams, 1995*; *Leibold, 1996*): what is the relationship between the amount of nutrient supplied and the resulting diversity of the community? Experimental studies of this question have mainly been performed in macroecological contexts (*Mittelbach et al., 2001*; *Waide et al., 1999*; *Adler et al., 2011*), though recently there has been increased focus on microbial systems (*Bienhold et al., 2012*; *Bernstein et al., 2017*). In microbial experiments, some evidence has supported the 'hump-shaped' unimodal trend predicted by many theories (*Kassen et al., 2000*). However, a meta-analysis by *Smith, 2007* found no consistent trend across microbial experiments. What we observe here is concordant with Smith's result: even in our highly simplified model, there is no general relationship between nutrient supply and diversity. Among the factors we find that influence this relationship are cross-feeding, relative enzyme budgets, differences in enzyme affinities, and differences in nutrient yields. That so much variation appears in a simple model suggests that real ecosystems are not likely to display a single universal relationship between nutrient supply and diversity.

## Results

We employ the serial dilution model depicted in *Figure 1A* (see *Appendix 1—table 1*). At the beginning of each batch ($t = 0$), we introduce the inoculum, defined as a collection of species $\{\sigma\}$ with initial biomass densities $\rho_\sigma(0)$ in the batch such that the total initial biomass density is $\rho_0 = \sum_\sigma \rho_\sigma(0)$. Together with the inoculum, we supply a nutrient bolus, defined as a mixture of $p$ nutrients each with concentration in the batch $c_i(0)$, $i = 1, \ldots, p$ such that the total nutrient concentration is $c_0 = \sum_{i=1}^p c_i(0)$ (we also consider the case of cycles of single nutrient boluses that approach a mixture distribution, *cf. Appendix 7—figure 1*). It is assumed that all nutrients are substitutable (i.e. all nutrients are members of a single limiting class of nutrients, e.g. nitrogen sources). For

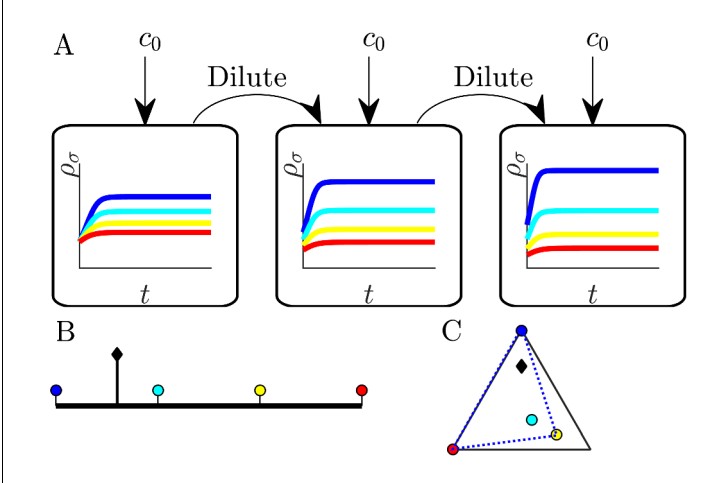

**Figure 1.** Illustration of serial dilution resource-competition model. (A) Serial dilution protocol. Each cycle of batch growth begins with a cellular biomass density $\rho_0$ and total nutrient concentration $c_0$. The system evolves according to *Equations 2-3* until nutrients are completely consumed. A sample of the total biomass is then used to inoculate the next batch again at density $\rho_0$. (B) Representation of particular enzyme-allocation strategies $\{\alpha_\sigma\}$ (colored circles) and nutrient supply composition $c_i/c_0$ (black diamond) on a 2-nutrient simplex, where the right endpoint corresponds to $c_1/c_0 = 1$. (C) Representation of particular strategies (circles) and nutrient supply (black diamond) on a 3-nutrient simplex. Dashed blue - the convex hull of the enzyme-allocation strategies. Here, the nutrient supply (black diamond) is inside the convex hull, implying coexistence of all species in the chemostat limit (see text).

simplicity, we assume ideal nutrient to biomass conversion, so that for a species to grow one unit of biomass density, it consumes one unit of nutrient concentration (we consider the case of nutrient-specific yields $Y_i$ in a later section). During each batch, the species biomass densities $\rho_\sigma(t)$ increase with time, starting at $t = 0$, and growth continues until the nutrients are fully depleted, $\sum_{i=1}^{p} c_i(\infty) \approx 0$ (we consider the case of incomplete depletion in *Appendix 7—figure 2*). Thus, at the end of a batch, the total biomass density of cells is $\sum_\sigma \rho_\sigma(\infty) = \rho_0 + c_0$. The next batch is then inoculated with a biomass density $\rho_0$ with a composition that reflects the relative abundance of each species in the total biomass at the end of the previous batch. This process is repeated until 'steady state' is reached, i.e. when the biomass composition at the beginning of each batch stops changing.

In the model, a species σ is defined by its unique enzyme strategy $\vec{\alpha}_\sigma = (\alpha_{\sigma,i}, \ldots, \alpha_{\sigma,p})$ which determines its ability to consume different nutrients. We assume that each species can consume multiple nutrients simultaneously, in line with the behavior of microbes at low nutrient concentrations (*Kovárová-Kovar and Egli, 1998*), though this assumption may not hold for all microbial species. Specifically, we assume that species σ consumes nutrient $i$ at a rate $j_{\sigma,i}$ (per unit biomass) that depends on nutrient availability $c_i$ and on its enzyme-allocation strategy $\alpha_{\sigma,i}$ according to

$$j_{\sigma,i} = \frac{c_i}{K_i + c_i} \alpha_{\sigma,i}. \tag{1}$$

For simplicity, we take all Monod constants to be equal, $K_i = K$ (a more general form of the

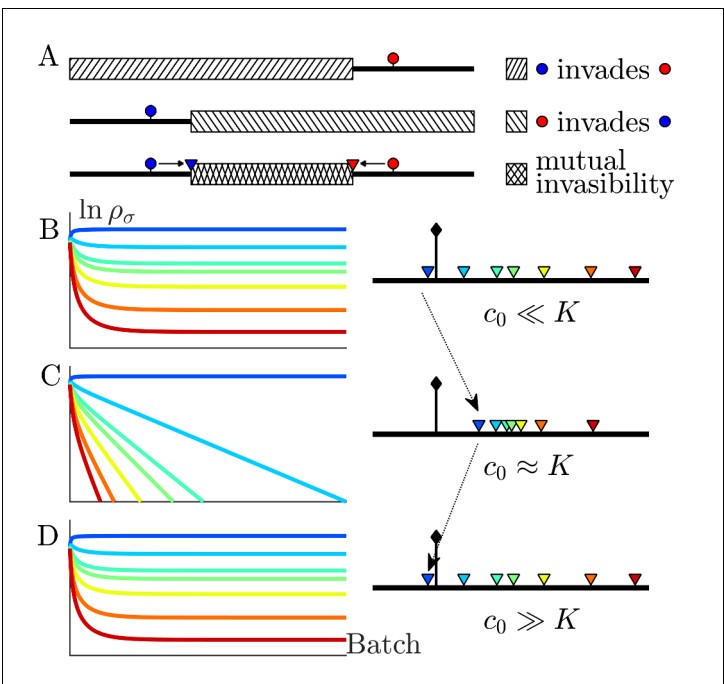

**Figure 2.** The nutrient bolus size $c_0$ affects the relative abundance of species and even their coexistence at steady state. (**A**) Schematic of the mutual invasibility condition for two species and two nutrients. Top: The red species can be invaded by any species with a strategy to its left if the supply lies in the region marked by the hatched rectangle. Middle: Similarly, showing the supplies for which blue can be invaded by any species with a strategy to its right. Bottom: The intersection defines a mutual invasibility region of supplies for which the two species red and blue will coexist. Triangles mark the boundaries of this coexistence region. (**B–D**) Example of the effect of $c_0$ on coexistence for more than two species: the approach to steady state, showing $\rho_\sigma$ versus batch number (left column) with the corresponding $c_0$-dependent remapping of coexistence boundaries (right column). (**B**) For the chemostat limit $c_0 \ll K$, where $K$ is the Monod constant for nutrient uptake, the triangles marking coexistence boundaries coincide with the species' strategies, $\alpha_\sigma$. (**C**) For $c_0 \approx K$ the triangles are remapped towards the center of the simplex compared to the strategies $\{\alpha_\sigma\}$. In this example the nutrient supply (black diamond) ends up outside the coexistence boundaries, so only one species survives. (**D**) For $c_0 \gg K$ the triangles again coincide with the strategies $\{\alpha_\sigma\}$, leading again to coexistence.

nutrient model is considered in a later section). During each batch, the dynamics of nutrient concentrations and biomass densities then follow from the rates $j_{\sigma,i}$ at which the species consume nutrients:

$$\frac{dc_i}{dt} = -\sum_{\sigma} \rho_{\sigma} j_{\sigma,i},$$

(2)

$$\frac{d\rho_{\sigma}}{dt} = \rho_{\sigma} \sum_{i} j_{\sigma,i}$$

(3)

Since the level of one enzyme inevitably comes at the expense of another, we model this trade-off via an approximately fixed total enzyme budget $E$. Formally, we take $\sum_i \alpha_{\sigma,i} = E + \varepsilon \xi_{\sigma}$, where $\xi_{\sigma}$ is a zero-mean and unit-variance Gaussian variable. Without loss of generality we take $E = 1$; initially we set $\varepsilon = 0$, which we call *exact* trade-offs. This allows us to visualize the strategies $\vec{\alpha}_{\sigma}$ as points on a simplex, depicted as colored circles embedded in: (i) the interval $[0,1]$ for two nutrients (*Figure 1B*), or (ii) a triangle for three nutrients (*Figure 1C*), etc. One can plot the nutrient bolus composition $c_i/c_0$ on the same simplex, as depicted by the black diamonds in *Figure 1B and C*. In what follows, we focus on the case of two nutrients, though the main results extend to an arbitrarily large number of nutrients.

## Connection between serial dilution and chemostat models

One can intuit that our serial dilution model at very low nutrient bolus size will mimic a chemostat. Adding a small nutrient bolus, letting it be consumed, then removing the additional biomass, and repeating is tantamount to operating a chemostat with a fixed nutrient supply and dilution rate. Indeed, the limit $c_0 \ll K$ yields the same steady state as a chemostat. Thus, our results for serial dilution include and generalize those obtained for a closely related chemostat model (*Posfai et al., 2017*).

For completeness, we now briefly describe the chemostat results from *Posfai et al., 2017*. In the presence of metabolic trade-offs, the chemostat can support a higher species diversity than prescribed by the competitive exclusion principle as we demonstrate theoretically in *Appendix* 4. Specifically, if the nutrient supply lies within the convex hull of the strategies on the simplex (visualized by stretching a rubber band around the outermost strategies, see *Figure 1B–C*), an arbitrarily large number of species can coexist at steady state. In the chemostat, such species coexistence is attained when the system organizes such that all nutrient levels are driven towards equality by consumption. Dynamically, if one nutrient level is high, the species that consume it increase in population, leading to faster consumption of that nutrient, thus acting to return the nutrients to equal steady-state levels. Such a self-organized neutral state is an attractor of the chemostat dynamics (*Posfai et al., 2017*) and, correspondingly, of the $c_0 \ll K$ limit of the serial dilution model. Note that the coexistence steady state is not a single fixed point, but rather a degenerate manifold of possible solutions (details in Appendix 4).

Thus, in the chemostat-limit of the cases shown in *Figure 1B and C* all the species will coexist. Conversely, if the supply lies outside the convex hull, (e.g., if we swapped the positions of the leftmost species and the supply in *Figure 1B*) the number of surviving species would be strictly less than the number of nutrients, consistent with competitive exclusion. To understand the convex-hull rule, note that a state of arbitrarily high coexistence can only occur if the chemostat self-organizes to a 'neutral' state in which the nutrient concentrations are all equal, and thus all strategies have the same growth rate. This state is achieved if and only if the total enzyme abundances lie along the same vector as the nutrient supply, which is achievable only if the supply lies within the convex hull of the strategies present.

As in the chemostat model, the serial dilution model can support either coexistence or competitive exclusion. However, if one chooses system parameters near the transition between these two states, it requires a very large number of batches for the simulation to reach steady state. This is a manifestation of the well-known phenomenon of *critical slowing down* (*Dakos and Bascompte, 2014*). Though in principle, critical slowing down is not a simulation artifact and could manifest in similar real-world systems, we expect that a variety of factors outside our modeling framework would preclude observation of this critical behavior.

We define the serial dilutions 'steady state' to be reached when the relative species abundances after the nutrients are depleted (time $t_f$ after starting the batch) scale with the relative abundances at the beginning of that batch (time 0), that is, $\rho_\sigma(t_f) = \frac{\rho_0 + c_0}{\rho_0}\rho_\sigma(0)$. We can expand the implicit equation for the steady state to first order in $c_0/K$ (details can be found in Appendix 4),

$$\frac{c_0}{\rho_0} = \sum_i \frac{\alpha_{\sigma,i} c_i(0)}{\sum_{\sigma'} \alpha_{\sigma',i} \rho_{\sigma'}(0)} + O\left(\frac{c_0}{K}\right)^2. \tag{4}$$

Dividing both sides by $t_f$ and defining,

$$\tilde{\delta} = \frac{c_0}{\rho_0 t_f}, \quad s_i = \frac{c_i(0)}{t_f}, \tag{5}$$

we reach the $c_0/K \ll 1$ steady-state condition for the serial dilution system:

$$\tilde{\delta} = \sum_i \frac{\alpha_{\sigma,i} s_i}{\sum_{\sigma'} \alpha_{\sigma',i} \rho_{\sigma'}(0)}. \tag{6}$$

Averaged over a batch, $s_i$ is the average rate that nutrient $i$ is supplied, and $\tilde{\delta}$ is the average rate that all the nutrients are supplied per unit inoculum biomass. In analogy to the chemostat model, one can think of $s_i$ as the rate nutrient $i$ is continuously supplied. Moreover, for a chemostat, the parameter $\tilde{\delta}$ which would be the rate all nutrients are continuously supplied per unit biomass, would need to equal the dilution rate of the chemostat, $\delta$, to maintain steady state. A detailed analysis of the effects of larger bolus size, to second order in $c_0/K$, can be found in Appendix 4.

## Effect of total nutrient bolus on coexistence

In the chemostat limit, increasing the nutrient supply rate simply proportionally increases the steady-state population abundances. However, away from this limit we find that the magnitude of the nutrient bolus can qualitatively affect the steady-state outcome of serial dilutions. To understand this effect, we first consider a simple case of two nutrients and two species as depicted in *Figure 2A*. The two species will coexist if each species is invasible by the other. In our example, we first determine the invasibility of species R (strategy indicated by red circle) by species with strategies lying to its left. To this end, we choose a nutrient supply and perform model serial dilutions until steady state is reached. For a particular finite bolus size, we find that for all supplies within the hatched region an infinitesimal inoculum of any species lying to the left of R will increase more than R during a batch, and therefore can invade R. Similarly, we determine the invasibility of species B (strategy indicated by blue circle) by any species with a strategy lying to its right, and find the second hatched region. The intersection of these hatched regions for which (1) B can invade R and (2) R can invade B is the supply interval of mutual invasibility where these two species will stably coexist. The coexistence interval is bounded by the red and blue triangles, and each of these coexistence boundaries is a unique property of its corresponding species. We call these species-specific boundaries *remapped* because they generally lie at different locations on the simplex than the strategies they originated from, with the extent of remapping depending on the nutrient bolus size. At a more technical level, the remapped boundary for a given species and bolus size is the nutrient supply for which, over the course of a batch, all nutrients are equally valuable and so, a species with any strategy can neutrally invade and persist. This equality of nutrient value is defined in terms of the Monod function integrals for each nutrient (for details see Appendix 3).

Since the remapped coexistence boundaries depend on the nutrient bolus size $c_0$, changing bolus size can qualitatively change the steady-state outcome of serial dilutions. *Figure 2B–D* depicts an example of how $c_0$ affects remapping, and the consequences for species coexistence. At low bolus size, $c_0 \ll K$, corresponding to the chemostat limit, *Figure 2B* (left) shows that all species present achieve steady-state coexistence. This follows because the nutrient supply (black diamond) lies inside the convex hull. When $c_0$ is increased to $c_0 \approx K$ (*Figure 2C*), the coexistence boundaries are remapped towards the center of the simplex (dashed arrow). In this example, the nutrient supply now lies outside the convex hull. This results in one winner species (the dark blue one nearest the supply), with all others decreasing exponentially from batch to batch. This loss of coexistence with increasing nutrient bolus size is reminiscent of Rosenzweig's 'paradox of enrichment' in predator-

prey systems (*Rosenzweig, 1971*). Strikingly, however, as bolus size is further increased to $c_0 \gg K$, the coexistence boundaries are remapped back to their original positions, so that the nutrient supply once again lies within the convex hull, and so steady-state coexistence of all species is recovered.

What causes the remapping of the coexistence boundaries inwards as $c_0/K \to 1$? Let us consider a single species growing on two nutrients supplied in the same proportion as its strategy (i.e. the fraction of Nutrient 1 is equal to $\alpha_{\sigma,1}$). At low $c_0/K$, this marks the remapped coexistence boundary and both nutrients will be equally valuable over the course of a batch. The balance is achieved because the nutrient with a higher initial abundance is more rapidly exhausted, while the nutrient with lower initial abundance is consumed more slowly and is therefore available for a longer span of time. At small $c_0/K$, the more rapid initial consumption of the more abundant nutrient does not influence the consumption rate of the less abundant nutrient because the bolus size is small relative to the initial population size, so the population does not grow substantially during the batch. This changes as $c_0/K$ increases: the rapid initial consumption of the more abundant nutrient leads to a substantial increase in the total population. The remaining low initial abundance nutrient is now consumed more quickly and is less available to an invader with a more balanced enzyme strategy. The nutrients are no longer equally valuable on average, and remedying this requires a more equally balanced nutrient bolus. Thus, the remapped coexistence boundary moves inwards (see *Appendix 7— figure 3*). In essence, as $c_0/K$ increases it is more difficult for the invader to grow because the resident gains an 'early-bird' advantage: its initial growth allows it to more effectively exhaust the nutrients.

Why does the coexistence boundary of a species map back to its original strategy in the limit of large bolus size, $c_0 \gg K$? In this limit, the nutrient uptake functions in *Equation 1* will be saturated during almost the entire period of a batch. Each species will therefore consume nutrients strictly in proportion to its strategy $\alpha_{\sigma,i}$. For the case of two nutrients (e.g., as shown in *Figure 1B*), if there is only a single species present then if the supply lies anywhere to the left of its strategy, at some time during the batch there will be some of Nutrient 2 remaining after the bulk of Nutrient 1 has been consumed. Thus a single species can be invaded by any strategy to its left, provided the supply also lies to its left. Similarly, a species can be invaded by any strategy to its right if the supply lies to its

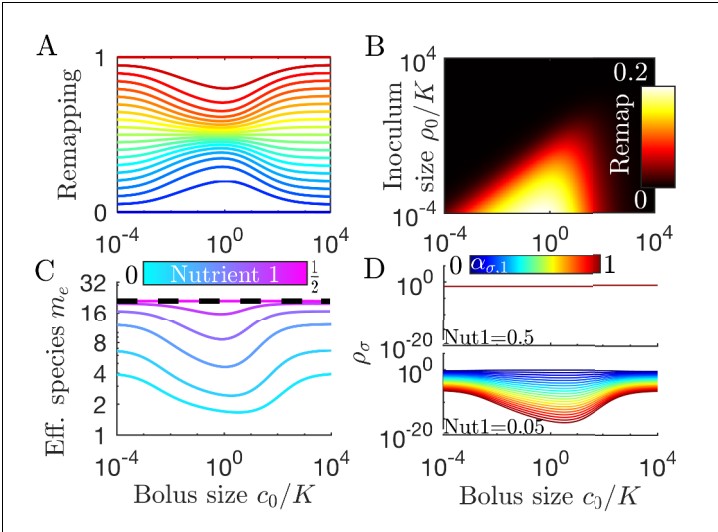

**Figure 3.** Remapping of strategies at finite nutrient supply generally reduces species diversity. (A) As shown for the case of two nutrients, the remapping of strategies (i.e., the shift of coexistence boundaries) is non-monotonic with nutrient bolus size $c_0$ (colors indicate 21 equally spaced strategies). (B) Heat map of the extent of remapping for strategy $(0.2, 0.8)$ as a function of nutrient bolus size $c_0/K$ and inoculum size $\rho_0/K$. (C) Steady-state effective number of species $m_e$ as a function of bolus size $c_0/K$ with equal initial inocula adding up to $\rho_0/K = 10^{-3}$; the same initial conditions apply for panels *C-D*. Colors correspond to different nutrient supply compositions $c_1/(c_1 + c_2)$. Dashed black line: maximum diversity (equal species abundances) is attained when nutrient composition is $(0.5, 0.5)$. (D) Steady-state species abundances $\{\rho_\sigma\}$ for nutrient composition $(0.5, 0.5)$ (top) and $(0.05, 0.95)$ (bottom).

right. This is exactly the condition for the coexistence boundary of a species to coincide with its actual strategy (details in *Appendix 5*).

We have rationalized coexistence in our serial dilution model in terms of mutual invasibility, but have not explicitly stated the condition for an arbitrary number of species to coexist in steady state. In the chemostat, all species coexist when the concentrations of all nutrients are equal, implying the same growth rate for all strategies. However, for serial dilutions the nutrient concentrations are generally not equal and are not even constant in time. Instead, it is the integrated growth contribution of every nutrient that must be equal to allow for arbitrary coexistence. In the case of equal enzyme budgets ($\varepsilon = 0$), this condition occurs when the time integrals of the nutrient Monod functions within a batch are all equal, that is,

$$I_i = \int_0^\infty \frac{c_i}{K_i + c_i} dt = \text{const.} \tag{7}$$

To understand this condition for coexistence beyond competitive exclusion, note that the instantaneous rate of growth of a species σ is $\sum_i \alpha_{\sigma,i} c_i/(K_i + c_i)$, so that the fold increase of a species during a batch is $\exp(\vec{\alpha}_\sigma \cdot \vec{I})$. This fold increase will be equal for all species if and only if *Equation 7* holds. When there are two nutrients, *Equation 7* holds at steady state whenever the supply is inside the convex hull of the coexistence boundaries of the species present (details in Appendix 3). For more nutrients, the corresponding condition is that the region of coexistence is bounded by contours that connect the outermost remapped nodes.

Given a fixed set of species and a choice of initial populations, repeating the growth-dilution batch procedure results in a steady state where the populations at the beginning of a batch do not change from batch to batch. The steady-state populations depend on the initial populations, with the set of all possible steady-state populations defining a coexistence manifold.

## Steady-state diversity

As is apparent in *Figure 2C*, not all strategies are remapped to the same extent. In *Figure 3A*, we plot the remapping of coexistence boundaries as a function of nutrient bolus $c_0$. Note that: (i) the specialists $(0, 1)$ and $(1, 0)$ and the perfect generalist $(0.5, 0.5)$ are not remapped at all; (ii) remapping is maximal for $c_0 \approx K$; (iii) there is no remapping in both the $c_0 \to 0$ and $c_0 \to \infty$ limits (see also *Appendix 7—figure 4*). The extent of remapping also depends on the inoculum size $\rho_0$ as shown in *Figure 3B*, which demonstrates that remapping is maximal for $\rho_0 \ll K$ and vanishes for $\rho_0 \gg K$.

How does nutrient bolus size influence steady-state species diversity? A useful summary statistic for quantifying diversity (*Jost, 2006*) is the effective number of species $m_e = e^S$ with the Shannon diversity $S = -\sum_\sigma P_\sigma \ln P_\sigma$ and $P_\sigma = \rho_\sigma^*(0)/\rho_0$, with $\rho_\sigma^*(0)$ the steady-state species abundances at the beginning of a batch. Diversity as measured by $m_e$ is shown in *Figure 3C* for six choices of nutrient bolus composition. Notably, if the two nutrients are supplied equally (top curve, magenta), $m_e$ is independent of $c_0$ and coincides with the maximal possible diversity (dashed black line), namely equal steady-state abundances of all species (*Figure 3D*, top). Conversely, if Nutrient 1 comprises only 5% of supplied nutrient (*Figure 3C*, bottom curve, cyan), the number of effective species $m_e$ is lower than maximal even in the chemostat-limit of small bolus sizes $c_0 \ll K$ and drops even further for $c_0 \approx K$. This loss of diversity is due to the dramatically lowered steady-state abundances of strategies that favor Nutrient 1 (*Figure 3D*, bottom). Two different effects underlie this change in community structure. The first is the early-bird effect described above: species specializing in more abundant nutrients gain a population advantage that allows them to rapidly consume less abundant nutrients that would otherwise support species with different enzyme specializations. The second effect is a well-known property of single nutrient competition and can be viewed as a 'single-nutrient' early-bird effect. In this case, species that are superior competitors for a nutrient gain an exponential population advantage over inferior competitors, increasing their share of total nutrient beyond the ratio of initial consumption rates. Both of these effects increase in strength with larger bolus size because the early-bird advantage increases as growth proceeds. The combination of these effects results in the species specialized in consuming the most abundant nutrients consuming a larger fraction of all nutrients. However, for very large bolus sizes, saturation of nutrient uptake rates mitigates these two effects, leading to a lack of remapping for $c_0 \gg K$ and diversity returning to its

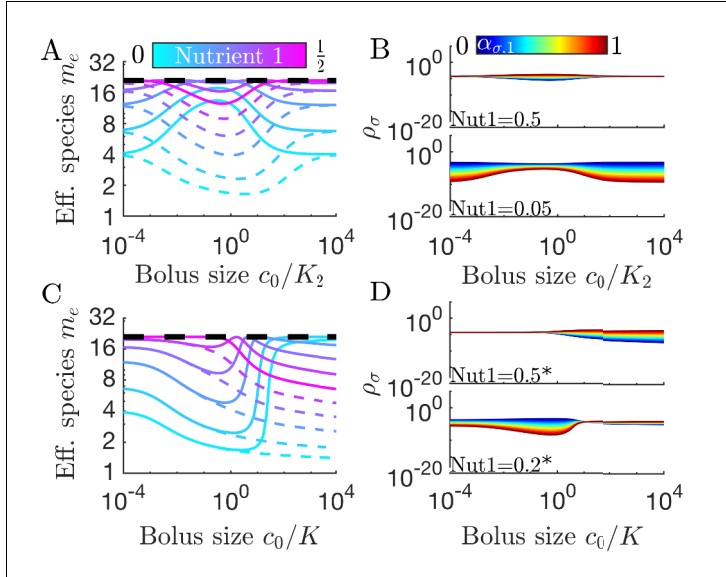

**Figure 4.** Differences in enzymes affinities, $K_i$, and nutrient yields, $Y_i$, lead to different relationships between diversity and bolus size. (A) Steady-state effective number of species $m_e$ as a function of bolus size $c_0/K_2$, as in **Figure 3C**, but with $K_1 = 10^{-3}$ and $K_2 = \rho_0 = 1$. Colors correspond to different nutrient supply compositions, solid curves for $c_1/c_0 \in [0, 0.5]$ and dashed curves for $c_1/c_0 \in [0.5, 1]$. Dashed black line: maximum diversity (equal species abundances) is no longer attained when nutrient composition is $(0.5, 0.5)$. (B) Steady-state species abundances $\{\rho_\sigma\}$ for nutrient composition $(0.5, 0.5)$ (top) and $(0.05, 0.95)$ (bottom), as in **Figure 3D**. (C) Steady-state effective number of species $m_e$ as a function of bolus size $c_0/K$, as in **Figure 3C**, but with $Y_1 = 10$ and $Y_2 = \rho_0 = 1$. Note that the nutrient compositions are normalized to yield such that $(0.5^*, 0.5^*)$ is actually $(0.5/Y_1, 0.5/Y_2)$. Colors the same as in A. (D) Steady-state species abundances $\{\rho_\sigma\}$ for nutrient composition $(0.5^*, 0.5^*)$ (top) and $(0.2^*, 0.8^*)$ (bottom).

chemostat-limit. Though here we focused on the case of two nutrients, these results extend to more nutrients (for three nutrients see **Appendix 7—figure 5**).

## Models with fewer simplifying assumptions

In this final Results section, we consider the effects of relaxing some of our simplifying assumptions. We first assess the effect or different enzyme affinities, $K_i \neq K$, and different nutrient yields $Y_i \neq Y$. This is followed by a model that allows cross-feeding of metabolites. Finally, we consider population bottlenecks and what happens when the fixed-enzyme-budget constraint is relaxed. We show that in

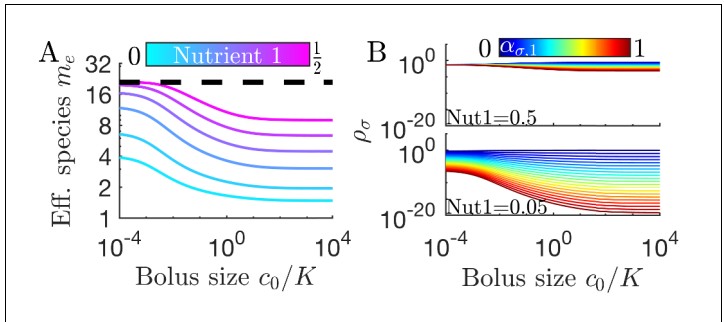

**Figure 5.** Cross-feeding alters the relationship between diversity and bolus size. (A) Steady-state effective number of species $m_e$ as a function of bolus size, as in **Figure 3C** but with two trophic layers, with Nutrient 1 a byproduct of metabolizing Nutrient 2. The byproduct fraction $\Gamma$ is chosen so that Nutrient 1 is produced at fractions according to the colorbar in A. (B) Steady-state species abundances $\{\rho_\sigma\}$ for nutrient composition $(0.5, 0.5)$ (top) and $(0.05, 0.95)$ (bottom), as in **Figure 3D**.

all these cases, the dependence on bolus size can be understood as manifestations of the early-bird effect.

## Unequal enzyme affinities and nutrient yields

We have thus far made the simplifying assumption that all enzymes have the same substrate affinity, such that $K_i \equiv K$. However, in nature different nutrients may have drastically different values of $K$. For example, the methanogen *Methanosarcina barkeri* has $K_i$ for the consumption of hydrogen and acetate that differ by approximately three orders of magnitude (*Robinson and Tiedje, 1984*; *Wandrey and Aivasidis, 1983*). How would such a large difference in the $K_i$ values impact diversity in our serial dilution ecosystem? In *Figure 4A* we show diversity as a function of bolus size for a system with a large difference in $K_i$ ($K_1 = 10^{-3}$, $K_2 = 1$). Since the symmetry between nutrients is broken by the unequal $K_i$, we now show the entire range of nutrient proportions, not just the first half. In the chemostat limit, the diversity values are similar to those found in the system with equal $K_i$. This makes sense: in a chemostat the nutrients with higher $K_i$ can accumulate to higher levels to compensate for their slow consumption, leaving the steady-state behavior unchanged. However, outside the chemostat regime, differences in the $K_i$ have a drastic effect: when the nutrient with the lower $K_i$ is scant in supply, diversity generally increases with increasing $c_0$, while the opposite occurs when the nutrient with the lower $K_i$ is higher in supply.

We can understand these $K_i$-driven shifts in the nutrient-diversity relationship as due to changes in the identity of the early bird. In a model with equal $K_i$, the identity of the early bird is determined by which nutrient is more abundant: if the two nutrients have equal $K_i$, a species can gain an early-bird advantage by preferentially consuming the more abundant nutrient. This changes if one nutrient has a much lower $K_i$ than the other. In this case it may be advantageous to preferentially consume the nutrient with the lower $K_i$, even if it is the less abundant nutrient. If the nutrient with the lower $K_i$ is also the more abundant nutrient, this will intensify the early-bird advantage. Why does this change in the early bird's identity change the form of the nutrient-diversity relationship? This change arises from a clash between optimal feeding behavior in the chemostat and seasonal regimes. In the chemostat, it is advantageous to focus on the most abundant nutrient, regardless of the value of $K_i$. Thus, in the chemostat limit, species focusing on the more abundant nutrient have an advantage. In the case of equal $K_i$ (or $K_i$ favoring the more abundant nutrient), this advantage is intensified by the early-bird effect, increasing the biomass of already abundant species and lowering diversity. By contrast, if the low abundance nutrient has a low $K_i$, the early-bird effect will have the opposite effect on diversity. Now the early-bird effect benefits species that were disadvantaged in the chemostat limit, leading to more equal abundances and higher diversity. This shift in abundances is shown in *Figure 4B*. The change in the identity of the early bird can also explain more complex relationships between diversity and bolus size (see *Appendix 7—figure 6*).

In addition to unequal enzyme affinities, it is possible for different nutrients to have different yields, $Y_i$. In *Figure 4C* we show the relationship between bolus and diversity for a system with $Y_1 = 10$ and $Y_2 = 1$. As expected, at low $c_0$ the diversity is similar to that in the case of equal $Y_i$. As $c_0/K$ increases, the diversity decreases initially and the symmetry-related bolus-composition cases (e.g. [0.2,0.8] and [0.8,0.2]) eventually diverge, with one's diversity rising and the other's continuing to fall. This behavior is explainable by the same logic as in the variable $K_i$ case: diversity rises or falls depending on whether the early-bird species was also favored in the chemostat limit. However, unlike the case of variable $K_i$, the diversity curves do not eventually return to the chemostat limit. Regardless of which nutrient the $Y_i$ favor, the diversity eventually begins decreasing monotonically as $c_0/K$ increases. This difference between the variable $K_i$ and variable $Y_i$ cases can be understood by considering what occurs when both nutrients are saturating. In the variable $K_i$ case, saturating nutrients are equal in value, implying a return to the chemostat limit as the early-bird effect weakens. In contrast, for variable $Y_i$, there remains a difference in the value of the two nutrients in the saturated regime, meaning that the early-bird effect will grow stronger and the early bird will take over the population. The beginning of this takeover can be seen at high bolus sizes in *Figure 4D*. Note that for both variable $K_i$ and variable $Y_i$, these trends are also reflected in the remapping (see *Appendix 7—figure 7*).

Despite the large variation in relationships between diversity and bolus size, these phenomena can all be understood as consequences of the early-bird effect. As the model becomes more

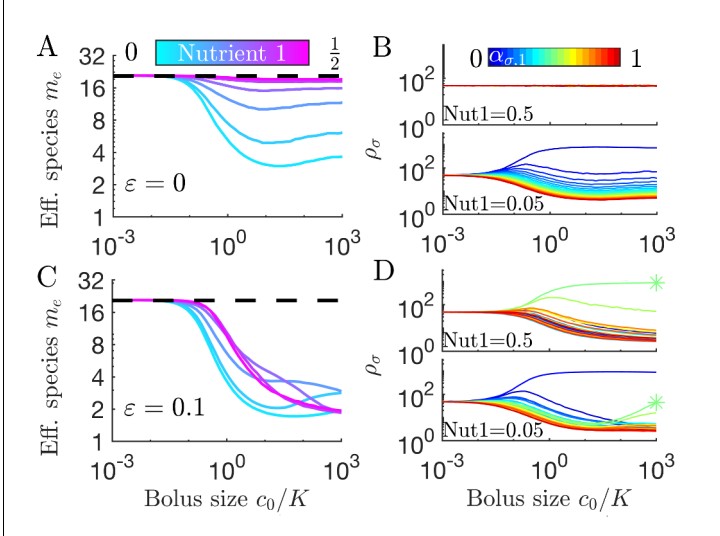

**Figure 6.** Diversity of small communities with migration. Each batch was inoculated with 1008 cells: 958 cells sampled without replacement from the previous batch, 50 cells sampled from 21 equally abundant, equally spaced strategies. (A) Effective number of species $m_e$ for different compositions of two nutrients (colors) as a function of nutrient bolus size $c_0/K$. (B) Average steady-state species abundances $\{\rho_\sigma\}$ for nutrient composition $(0.5, 0.5)$ (top) and $(0.05, 0.95)$ (bottom). (C) As A, but with random species-specific total enzyme budget specified by $\varepsilon = 0.1$. (D) As B but with species-specific enzyme budgets from C. Asterisk (*) indicates the species with the largest enzyme budget.

complex there are additional factors to consider in determining which nutrient will provide an early-bird advantage, but the fundamental mechanism of exploiting early growth advantages remains.

## Cross-feeding

It is possible to extend *Equations 2 and 3* beyond a single trophic layer, allowing for consumption of metabolic byproducts. This is a form of cross-feeding, which has generally been found to promote diversity (*Goyal and Maslov, 2018*) and stable community structure (*Goldford et al., 2018*). Here, cross-feeding is introduced through the byproduct matrix $\Gamma_{i,i'}^\sigma$, which converts the consumption of nutrient $i'$ to production of nutrient $i$ such that,

$$\frac{dc_i}{dt} = -\sum_\sigma \rho_\sigma \left( j_{\sigma,i} - \sum_{i'} \Gamma_{i,i'}^\sigma j_{\sigma,i'} \right). \tag{8}$$

In this framework, nutrient $i'$ is converted to nutrient $i$ at no extra enzymatic cost, meaning that nutrient $i$ is simply a byproduct whenever nutrient $i'$ is consumed for growth (it would be straightforward to modify this framework so that nutrient conversion can be carried out independently from growth). We focus on the simplest case: initially supplying only Nutrient 2, with Nutrient 1 solely derived as a metabolic byproduct via $\Gamma_{i,i'}^\sigma = \begin{pmatrix} 0 & \Gamma \\ 0 & 0 \end{pmatrix}$ for all species. When $\Gamma = 1$, upon consumption Nutrient 2 is perfectly converted to Nutrient 1, leading to an equal total supply of the two nutrients. More generally, $\int_0^\infty \sum_\sigma \rho_\sigma j_{\sigma,1} \, dt = \Gamma c_2(0)$ which allows a direct comparison between the unitrophic and bitrophic regimes: starting with $c_2(0)$ results in $(\Gamma + 1)c_2(0)$ total nutrient, and hence the Nutrient 1 fraction is $\frac{\Gamma}{1+\Gamma}$ of the total.

How does cross-feeding influence diversity in our serial dilution model? In *Figure 5A* we compare bitrophic diversity for six values of $\Gamma$ to their unitrophic equivalents (in *Figure 3C*). We note that: (i) bitrophy still supports diversity greater than the competitive-exclusion limit; (ii) in the chemostat regime, $c_0 \ll K$, the unitrophic and bitrophic schemes have identical values of $m_e$, and these drop as $c_0 \to K$; (iii) but for bitrophy the $m_e$ does not recover for $c_0 \gg K$; (iv) even when the total supply of both nutrients is equal ($\Gamma = 1$), bitrophy leads to lower than maximal $m_e$ outside the chemostat limit.

These features are clarified in *Figure 5B*, which shows steady-state species abundances for $\Gamma$ values leading to a total Nutrient 1 supply fraction of 0.5 and 0.05, and highlights the lower diversity for bitrophy compared to unitrophy for large nutrient bolus size. This difference is due to an early-bird effect: the species consuming supplied nutrient early in the batch can build a sizable population before the competing species that rely on its byproduct. The early-bird population then outcompetes the others for byproduct consumption. As such, this effect increases with $c_0/\rho_0$. The effect also becomes stronger at low $c_0/K$ (with constant $c_0/\rho_0$), since this allows the early-bird species more time to grow before the byproduct accumulates to high enough levels to be significantly consumed (*Appendix 7—figure 8*). Note that this effect is dependent on metabolite byproducts being also consumed by their producer. If the species in each trophic layer are single-nutrient specialists, then changes in $c_0/K$ have no impact on community diversity.

The behavior of the model with cross-feeding shows that the early-bird effect extends beyond simple metabolic trade-offs. More broadly, when species compete for multiple resources that are supplied in batches, a species' survival depends on more than its ability to efficiently consume nutrients. An early-bird species, being more specialized in consuming the nutrients that are initially more abundant, gains a population advantage early in the batch. This population advantage may allow the early-bird species to out-compete other species even when consuming nutrients it is not specialized to consume. Despite its consumption inefficiencies, through sheer numbers the early-bird species can consume more of the remaining nutrients than its more specialized competitors.

## Population bottlenecks

So far we have considered deterministic dynamics, which is appropriate for large populations. In natural settings, however, there are often small semi-isolated communities. For these communities, fluctuations can play an important role. In particular, population bottlenecks can lead to large demographic changes (*Abel et al., 2015*). In our model, how does the nutrient supply affect diversity in such communities? To address this question, we applied discrete sampling of a finite population when diluting from one batch to the next (see Appendix 1). With this protocol, an 'extinction' occurs when sampling yields zero individuals of a species. For a long enough series of dilutions such extinctions would ultimately lead to near-complete loss of diversity. For small real-world populations, however, diversity may be maintained by migration. To model such migration we augmented the population at each dilution with a 'spike-in' from a global pool of species, in the spirit of MacArthur's theory of island biogeography (*MacArthur and Wilson, 2001*). Specifically, in the spike-in procedure, to prevent extinctions caused by sampling fluctuations, every new batch is inoculated with a small number of the original, founder species.

In *Figure 6A* we show results of spike-in serial dilutions for a population bottleneck of 1008 cells. 95% of these cells are sampled from the previous batch, while 5% are sampled from a global pool, with equal abundances of 21 equally spaced strategies (*cf. Figure 3A*). The resulting $m_e$ vs. $c_0$ curves have maximal $m_e$ for all six nutrient fractions in the regime $c_0 \ll K$ where the 5% spike-in dominates sampling noise. As expected, for a balanced nutrient supply at any $c_0$, all species have the same average abundance (*Figure 6B* top). By contrast, when Nutrient 1's fraction is low (*Figure 6A* cyan and *6B* bottom), increasing $c_0$ increases the abundance gaps between the species, reflecting the uneven competition for Nutrient 2. Overall, the spike-in protocol leads to higher diversity at low $c_0$ than the deterministic case (starting from equal species abundances but with no spike-in, *Figure 3C*). For large $c_0$, the $m_e$ vs. $c_0$ curves for these two protocols are indistinguishable. The only noticeable difference is that the spike-in maintains a higher level of the least competitive strains, but since these abundances are still low, this difference in not reflected in the $m_e$ values.

### Unequal enzyme budgets

While we have assumed exact trade-offs to achieve diversity within a resource-competition model, the trade-offs present among real microorganisms will not be exact. For the serial dilution protocol with spike-ins, diversity is maintained by migration and so it is possible to relax the constraint of exact trade-offs. How does diversity depend on the nutrient supply if we allow species to have different enzyme budgets? We implemented random differences in species enzyme budgets by setting $\varepsilon = 0.1$, that is, a standard deviation of 10%, and plotted effective number of species $m_e$ in *Figure 6C*. As in the $\varepsilon = 0$ limit (*Figure 6A*), at sufficiently small $c_0$ the spike-in procedure dominates

both sampling noise and differential growth rates due to unequal enzyme budgets. Raising $c_0$ leads to a drop in $m_e$ (albeit still above the competitive-exclusion limit). Examining the species abundances in *Figure 6D*, we note that differences in enzyme budget establish a fitness hierarchy even when nutrient fractions are equal (top), with those species with the highest budgets increasing in relative abundance as $c_0$ increases. The asterisk (*) marks the species with the highest total enzyme budget, which becomes the most abundant for $c_0 \gg K$. Reducing Nutrient 1's fraction to 0.05 results in a shifting abundance hierarchy (*Figure 6D*, bottom): at low $c_0$ the highest abundance species is the one that consumes only Nutrient 2, as in the equivalent $\varepsilon = 0$ case. However, increasing $c_0$ results in increased abundance for the species with the highest enzyme budget – which would ultimately lead to its domination for sufficiently large $c_0$. This increasing dominance of the species with the highest enzyme budget is another manifestation of the early-bird effect: as the amount of growth in a batch increases, the advantage of a larger enzyme budget further compounds. In short, for spike-in serial dilutions the influence of unequal enzyme budgets depends on the nutrient supply, such that the species with the largest budgets dominate for large, unbiased supplies.

## Discussion

Natural ecosystems experience variations in the timing and magnitude of nutrient supply, and the impact of these variations on species diversity is not fully understood (*Smith, 2011*; *Smith, 2007*). To explore the impact of variable nutrient supply, we modeled resource competition in a serial dilution framework and analyzed the model's steady states. We found that variable nutrient supply still allows for the high diversity seen in the continuous supply ('chemostat') version of the model. Indeed, the serial dilution steady state mimics that of a chemostat when the amount of nutrients supplied in each batch is small. Surprisingly, however, supplying the nutrients as a bolus led to a dependence of diversity on the amount of supplied nutrients.

In contrast to existing literature on seasonality, we find that environmental fluctuations can both weaken and strengthen coexistence in this model. This occurs as the result of an 'early-bird' effect associated with supplying nutrients as large seasonal boluses instead of continuously. Some species can capitalize on rapid initial growth on an abundant nutrient to reach a large population size, which then allows them to deplete the remaining nutrients at the expense of their competitors. This early-bird effect can both restrict and expand the range of environments in which communities can self-organize to a neutral state. We show that even when metabolic trade-offs are combined with stabilizing mechanisms, the impact of the early-bird effect remains. For example, in the case of cross-feeding, the community diversity falls as a function of $c_0/K$ due to the early-bird advantages gained by species at higher trophic levels.

While the idea of species gaining early advantages has been explored, such as in the literature on founder effects and speciation (*Barton and Charlesworth, 1984*; *Brown,, 1957*), to the best of our knowledge this is the first demonstration of the influence of the early-bird effect on the diversity of seasonal ecosystems. We believe that this effect will occur in a variety of such ecosystems, as its only fundamental requirement is competition for multiple nutrients that are supplied in a time-dependent manner. Interestingly, while the early-bird effect plays a large role in our model, it is not the only bolus-dependent effect that influences diversity. We also observe another effect that can be viewed as a 'single-nutrient' version of the early-bird effect. This effect arises from a well-studied property of competition: as growth proceeds, a superior competitor for a nutrient gains an exponential advantage over inferior competitors for that nutrient. Like the early-bird effect, this shifts the system's biomass towards species more specialized in initially abundant nutrients, particularly for large but non-saturating nutrient bolus sizes. This single-nutrient effect can co-occur with the early-bird effect, for example in competition for an abundant nutrient between two early-bird species.

The form of seasonality we explore in this manuscript, where mixed boluses are supplied periodically, is only one possible form of seasonal nutrient supply. The impact of the early-bird effect and single-nutrient competition will likely differ between different forms of seasonality. For example, we show in *Appendix 7—figure 1* that supplying cycles of single nutrient boluses that approach an equal distribution of nutrients results in lower diversity than supplying mixed equal nutrient boluses. While this form of seasonality differs from the one we characterized, we can still understand the loss of diversity as arising from the single-nutrient competition effect initially observed in our mixed-bolus

models. We expect the principles gleaned from our models to be of use in understanding diversity in a variety of seasonal ecosystems.

Finding a general relation between the amount of nutrient supplied to a community and its diversity is a long-standing goal of theoretical ecology (*Tilman, 1982*; *Abrams, 1995*; *Leibold, 1996*). We found that in our model the form of the nutrient-diversity relation (NDR) can change based on model details. The model has two regimes: a low diversity and a high diversity regime. The former satisfies competitive exclusion (no more species coexisting than resources), whereas the latter exceeds competitive exclusion and occurs when the nutrient supply lies within the convex hull of the remapped metabolic strategies present (*Posfai et al., 2017*). At the bifurcation point between the two regimes, we observe critical slowing down in that the number of dilutions required to reach steady state diverges.

In the high diversity regime, the NDR can take several forms, resulting from the interplay of the early-bird effect and other mechanisms. Even with a single trophic layer, the NDR can be U-shaped, hump-shaped, monotonically decreasing, or have multiple peaks. These trends can then be further modified by the addition of more trophic layers, differences in enzyme budgets, etc.

Experimental studies that characterize the NDRs of microbial ecosystems have reached similarly variable conclusions. For example, one work studying bacterial communities in Arctic deep-sea sediments found an increasing trend between energy input and richness (*Bienhold et al., 2012*), while a study on photosynthetic microbial mats found a negative relationship between energy input and richness (*Bernstein et al., 2017*). A meta-analysis of aquatic microbial ecosystems found examples of both monotonic and non-monotonic NDRs, with no single form dominating (*Smith, 2007*). Our theoretical results, together with these experimental findings, indicate that there may be no single universal NDR in microbial ecosystems. This conclusion suggests that the best approach for characterizing the NDR of a given ecosystem is not to apply a one-size-fits-all theory, but to analyze the role of different factors such as cross-feeding, trade-offs, and immigration in determining that particular ecosystem's NDR. While we have focused on microbial systems, the absence of a universal NDR is consistent with results from recent work in plants (*Adler et al., 2011*).

We found that the stringency of metabolic trade-offs has a large impact on community diversity. We imposed a metabolic enzyme budget on each species to reflect the reality that microbial cells have a finite capacity to synthesize proteins and must carefully apportion their proteome (*Basan et al., 2015*). However, while it is true that microbes have limited biosynthetic capacity, it is unclear how strict are the resulting trade-offs. For this reason, we characterized versions of the model with both exact and inexact trade-offs. Our results show that the form of an ecosystem's NDR can depend on the stringency of metabolic trade-offs. This finding is not exclusive to the serial dilution model. The stringency of trade-offs was also important in the original chemostat setting: in a birth-death-immigration framework, small violations of the enzyme budget still allowed for high levels of coexistence, but large violations disrupted coexistence (*Posfai et al., 2017*). These results suggest that an experimental characterization of the stringency of metabolic trade-offs among microbes would provide a valuable ecological parameter. Note that metabolic trade-offs are only one of the many types of trade-offs microbes are subject to; other types of trade-offs, such as constraints between biomass yield and growth rate (*Wortel et al., 2018*), may also shape a community's NDR.

In constructing a model, we made a number of assumptions about the way in which microbes consume and utilize nutrients. Some of these assumptions do not apply to all microbial communities, and the impact of relaxing these assumptions can affect the NDR. For example, we mostly focused on communities where all nutrients are equally valuable (i.e. $Y_i = Y_j \; \forall i,j$). However, biomass yields can vary between nutrients and between species, which we explored in *Figure 4C–D*. Notably, unequal yields create differences between nutrients even in the saturating regime ($c_0 \gg K$), leading to a departure from the chemostat limit at large nutrient boluses. Coexistence in the serial dilution model is robust to varying yield, as long as all species have the same yield on a given nutrient. The scenario where species have different biomass yields on the same nutrient is conceptually similar to the case of inexact trade-offs, since some species will have a strict advantage over others. Thus, it is likely that these unequal yields between species will lead to a reduction in community diversity. However, varying the yield in this manner also allows for the inclusion of new trade-offs that may impact diversity, such as the aforementioned trade-off between yield and growth rate (*Wortel et al., 2018*). We also explored the effects of unequal Monod constants for different nutrients (*cf. Figure 4A–B*). We found that if a low-abundance nutrient also has a low $K_i$, the early-bird effect favors species that

were disadvantaged in the chemostat limit, thus reversing the equal-$K_i$ NDR and leading to hump-shaped NDR curves. Indeed, large differences in $K_i$ values can lead to a multi-peaked NDR as shown in *Appendix 7—figure 6*.

Our model assumes that all nutrients are substitutable (i.e. only one of the multiple nutrients is required for growth). In real ecosystems, microbes can require multiple complementary nutrients to grow, e.g. sources of carbon, nitrogen, and phosphorus. In cases where one class of complementary nutrient is strongly limiting, a model with both complementary and substitutable resources would essentially reduce to the current model of only substitutable resources. This case is likely the more common one, e.g. as many soils are carbon limited (*Aldén et al., 2001*; *Demoling et al., 2007*). However, in cases where no single nutrient is strongly limiting, the presence of complementary nutrients would possibly lead to different NDRs, which will be an interesting direction for future study.

Our modeling predictions, e.g. the convex hull condition and the changes in diversity due to the early-bird effect, are in principle testable. To connect our modeling assumptions to real microbial systems, we compare our growth model of substitutable and simultaneous nutrient consumption to previously published experimental data from *Escherichia coli* growing in batch and chemostat conditions. We find that our modeling assumptions are consistent with both datasets and outline potential future experiments to test the model's multispecies predictions, detailed in *Appendix 6*. As is apparent in *Appendix 6—figure 1*, the growth dynamics of *E. coli* at low nutrient levels is well described by our modeling framework. The experiments we compared were performed with the same strain of *E. coli*, meaning that inclusion of different microbes would be needed to test the multispecies predictions. To determine the strategies of other microbes, including other strains of *E. coli*, the most practical approach would likely be batch culturing. Once strains with different strategies have been identified, nutrient-diversity relationships could then be obtained by competing strains in serial dilution culture and measuring the community diversity (e.g. via fluorescent tags or by 16S rRNA sequencing) as a function of the total concentration of multiple, substitutable nutrients provided at the start of each batch.

## Additional information

### Funding

| Funder | Grant reference number | Author |
|---|---|---|
| National Institutes of Health | GM082938 | Amir Erez<br>Yigal Meir<br>Ned S Wingreen |
| National Science Foundation | DGE-1656466 | Jaime G Lopez |
| National Science Foundation | PHY-1734030 | Amir Erez<br>Jaime G Lopez<br>Benjamin G Weiner<br>Yigal Meir<br>Ned S Wingreen |
| National Science Foundation | PHY-1748958 | Yigal Meir<br>Ned S Wingreen |
| National Institutes of Health | R25GM067110 | Yigal Meir<br>Ned S Wingreen |
| Gordon and Betty Moore Foundation | 2919.01 | Yigal Meir<br>Ned S Wingreen |

The funders had no role in study design, data collection and interpretation, or the decision to submit the work for publication.

### Author contributions

Amir Erez, Jaime G Lopez, Conceptualization, Data curation, Software, Formal analysis, Validation, Investigation, Visualization, Methodology, Writing - original draft, Writing - review and editing; Benjamin G Weiner, Conceptualization, Investigation; Yigal Meir, Conceptualization, Formal analysis,

Investigation; Ned S Wingreen, Conceptualization, Resources, Formal analysis, Supervision, Funding acquisition, Investigation, Methodology, Writing - original draft, Project administration, Writing - review and editing

### Author ORCIDs
Amir Erez (ID) https://orcid.org/0000-0002-2320-4984
Jaime G Lopez (ID) https://orcid.org/0000-0003-1647-5898
Benjamin G Weiner (ID) http://orcid.org/0000-0002-1995-8660
Ned S Wingreen (ID) https://orcid.org/0000-0001-7384-2821

### Decision letter and Author response
Decision letter https://doi.org/10.7554/eLife.57790.sa1
Author response https://doi.org/10.7554/eLife.57790.sa2

## Additional files
### Supplementary files
• Transparent reporting form

### Data availability
No datasets were generated in this work. All code and data used in this manuscript available at: https://github.com/AmirErez/SeasonalEcosystem (copy archived at https://github.com/elifesciences-publications/SeasonalEcosystem).

The following datasets were generated:

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

# Appendix 1

## Methods

This section describes the simulation methods used in this manuscript. All code and data used in this manuscript can be found at https://github.com/AmirErez/SeasonalEcosystem (*Erez, 2020*; copy archived at https://github.com/elifesciences-publications/SeasonalEcosystem).

## Deterministic dynamics

We numerically solve the ODEs within each batch using a custom MATLAB-coded fourth-order Runge-Kutta solver with adaptive step size. Step size at a given time step is chosen such that the relative change of all state variables is below a predetermined threshold.

## Population bottleneck sampling

We implement discrete sampling when diluting from one batch to the next by picking without replacement $\rho_0$ individuals from a total end-of-batch population of $\rho_0 + c_0$. If there are non-integer populations at the end of a batch (as can occur with deterministic dynamics), they are rounded up if $\rho_\sigma - \text{floor}(\rho_\sigma) > U(0,1)$ where floor rounds down to the nearest integer and $U(0,1)$ is a uniform random variable between 0 and 1. For all simulations with stochastic bottlenecks, we allow the simulation to equilibrate for 10,000 dilutions and average over 10,000 further dilutions.

**Appendix 1—table 1.** Annotation glossary.

| Symbol | Description |
| --- | --- |
| $t$ | Time measured from the beginning of a batch |
| $p$ | Number of nutrients |
| $m$ | Number of species introduced at time $t = 0$ |
| $m_e$ | Effective number of species at steady state |
| $i$ | $(1...p)$ Latin index enumerating nutrients |
| $c_i(t)$ | Time dependent concentration of nutrient $i$ |
| $c_0$ | $\sum_{i=1}^{p} c_i(0)$; total nutrient concentration at time $t = 0$ |
| $K_i$ | Monod half-velocity constant for nutrient $i$ |
| $l_i$ | $\int_0^\infty \frac{c_i}{K_i + c_i} \, dt$; nutrient Monod function time integral |
| $Y_i$ | Biomass yield on nutrient $i$ |
| $\Delta$ | The fraction of nutrient remaining at the end of the batch |
| $s_i$ | Average rate that nutrient $i$ is continuously supplied at the chemostat limit |
| $\tilde{\delta}$ | Average rate all nutrients are continuously supplied at the chemostat limit |
| $\delta$ | Continuous chemostat dilution rate at the chemostat limit |
| $\sigma, \sigma', ...$ | $(1...m)$ Greek indices enumerating species |
| $\rho_\sigma(t)$ | Species $\sigma$ biomass density at time $t$ since a start of the batch |
| $\vec{\alpha}_\sigma$ | $(\alpha_{\sigma,1}, ..., \alpha_{\sigma,p})$; enzyme allocation strategy for species $\sigma$ |
| $\varepsilon$ | Standard deviation in enzyme budget |
| $E$ | $E = \sum_i \alpha_{\sigma,i} = 1$ for $\varepsilon = 0$; enzyme budget |
| $\Gamma_{i,i'}^\sigma$ | Byproduct matrix converting nutrient $i'$ to nutrient $i$ |
| $j_{\sigma,i}$ | Nutrient $i$ consumption rate by species $\sigma$ |

## Appendix 2

### General form of the model

The most general form of the model considered in this manuscript includes variable nutrient yield $Y_i$, Monod half-velocity constant $\tilde{K}_i$, and enzyme cost $w_i$:

$$\frac{d\rho_\sigma}{dt} = \sum_i \tilde{Y}_i \tilde{\alpha}_{\sigma i} \rho_\sigma \frac{\tilde{c}_i}{\tilde{K}_i + \tilde{c}_i} \tag{9}$$

$$\frac{d\tilde{c}_i}{dt} = -\sum_\sigma \tilde{\alpha}_{\sigma i} \rho_\sigma \frac{\tilde{c}_i}{\tilde{K}_i + \tilde{c}_i} \tag{10}$$

$$E = \sum_i w_i \tilde{\alpha}_{\sigma i}. \tag{11}$$

The enzyme costs $w_i$ and total enzyme budget $E$ of the original equations can be removed by rescaling the strategies and nutrient concentrations such that $\tilde{\alpha}_{\sigma i} = (E/w_i)\alpha_{\sigma i}$ and $\tilde{c}_i = (E/w_i)c_i$. This rescaling leads to a new effective Monod half-velocity constant and yield such that $\tilde{K}_i = (E/w_i)K_i$ and $\tilde{Y}_i = (w_i/E)Y_i$. The simplified equations are therefore:

$$\frac{d\rho_\sigma}{dt} = \sum_i Y_i \alpha_{\sigma i} \rho_\sigma \frac{c_i}{K_i + c_i} \tag{12}$$

$$\frac{dc_i}{dt} = -\sum_\sigma \alpha_{\sigma i} \rho_\sigma \frac{c_i}{K_i + c_i} \tag{13}$$

$$1 = \sum_i \alpha_{\sigma i}. \tag{14}$$

A further rescaling with $c'_i = c_i Y_i$, $K'_i = K_i Y_i$, and $\alpha'_{\sigma i} = \alpha_{\sigma i} Y_i$ reveals the impact of $Y_i$:

$$\frac{d\rho_\sigma}{dt} = \sum_i \alpha'_{\sigma i} \rho_\sigma \frac{c'_i}{K'_i + c'_i} \tag{15}$$

$$\frac{dc'_i}{dt} = -\sum_\sigma \alpha'_{\sigma i} \rho_\sigma \frac{c'_i}{K'_i + c'_i} \tag{16}$$

$$1 = \sum_i \left( \frac{\alpha'_{\sigma i}}{Y_i} \right). \tag{17}$$

## Appendix 3

### Mutual invasibility condition for coexistence beyond competitive exclusion

In our model, coexistence of an unlimited number of species can be traced back to the conditions for the coexistence of a smaller number of species. This is because in a system with $p$ nutrients, once $p$ species coexist they create an environment where all nutrients are equally valuable and all species can coexist. For example, understanding the conditions for unlimited coexistence in two nutrient competition requires us to examine the conditions that allow two species to coexist. In order for two species to coexist, they must be able to invade each other. This means that in an environment dominated by Species 1, Species 2 will have higher fitness and vice versa.

Under what nutrient supplies, $c_i(0)/c_0$, can two species invade each other? In the chemostat version of the model, these invasibility conditions are simple to determine. Consider two species where $\vec{\alpha}_1$ is to the left of $\vec{\alpha}_2$ on the 1-simplex. Species 2 can invade Species 1 if the nutrient supply is to the right of $\vec{\alpha}_1$. Species 1 can invade Species 2 if the nutrient supply is to the left of $\vec{\alpha}_2$. Therefore, the two species can mutually invade and coexist if and only if the nutrient supply lies between $\vec{\alpha}_1$ and $\vec{\alpha}_2$. This is precisely the convex hull condition, with no remapping.

For the same pair of species, how do we determine the nutrient supplies for which Species 1 can be invaded by Species 2 in the serial dilution version of the model? The fitness of a species in this model is the growth exponent $\sum_i \alpha_{\sigma,i} I_i$, meaning that Species 2 can invade Species 1 at nutrient supplies where Species 1 creates an environment such that $I_2 > I_1$. The nutrient supply at which Species 1 creates an environment where $I_1 = I_2$ therefore bounds the region of nutrient supplies for which Species 2 can invade. By the same logic, the border for the region where Species 1 can invade Species 2 is the nutrient supply at which Species 2 creates an environment where $I_1 = I_2$. Therefore, the mutual invasibility region is now defined by the nutrient supplies where each species growing in isolation creates an environment where $I_1 = I_2$. These points are what we refer to as the "remapped coexistence boundaries' and, unlike in the chemostat version of the model, these generally do not correspond to the species' strategies.

## Appendix 4

### Perturbation theory for $c_0/K \ll 1$

In the main text, we provide an explanation why in the limit of small nutrient bolus size $c_0/K$, the serial dilution model effectively becomes a chemostat. In this section, we prove this chemostat limit using a perturbation expansion to first order in $c_0/K$. Essentially, the Monod constant $K$ acts as the unit of nutrient and biomass in the system, which are measured in dimensionless units $c_0/K$ and $\rho_0/K$, respectively. Alternatively, one might choose to expand around a third ratio, $c_0/\rho_0$, which would be useful to model extremely nutrient-dilute conditions as found in some marine microbial ecosystems.

We define a perturbation expansion with respect to the small parameter $\phi = c_0/K$,

$$
\begin{aligned}
\rho_\sigma(t) &= \rho_\sigma(0) + \phi \rho_\sigma^{(1)}(t) + \phi^2 \rho_\sigma^{(2)}(t) + \dots \\
c_i(t) &= \phi c_i^{(1)}(t) + \phi^2 c_i^{(2)}(t) + \dots \\
\rho_\sigma^{(k>0)}(0) &= 0 \\
c_i^{(k>1)}(0) &= 0.
\end{aligned}
\tag{18}
$$

We note that at $O(1)$ we have $\rho_\sigma(t) = \rho_\sigma(0)$ and $c_i(t) = 0$ as expected. We begin by expanding the Monod function,

$$
\begin{aligned}
\frac{c_i}{c_i + K} &\approx \frac{c_i}{K} - \left(\frac{c_i}{K}\right)^2 \\
&\approx \left(\frac{\phi c_i^{(1)} + \phi^2 c_i^{(2)} + \dots}{K}\right) - \left(\frac{\phi c_i^{(1)} + \phi^2 c_i^{(2)} + \dots}{K}\right)^2 \\
&\approx \phi \left(\frac{c_i^{(1)}}{K}\right) + \phi^2 \left(\left(\frac{c_i^{(2)}}{K}\right) - \left(\frac{c_i^{(1)}}{K}\right)^2\right) + O(\phi^3).
\end{aligned}
\tag{19}
$$

Accordingly, in the kinetic equation for $c_i$,

$$
\dot{c}_i = -\frac{c_i}{K + c_i} \sum_\sigma \alpha_{\sigma,i} \rho_\sigma,
\tag{20}
$$

substituting the expansion in *Equation 19* and keeping the leading order, $c_i^{(1)}$, gives,

$$
\begin{aligned}
\dot{c}_i^{(1)} &= -c_i^{(1)} \underbrace{\sum_\sigma \frac{\alpha_{\sigma,i} \rho_\sigma(0)}{K}}_{\gamma_i} \\
\implies c_i^{(1)} &= c_i^{(1)}(0) e^{-\gamma_i t}.
\end{aligned}
\tag{21}
$$

We next solve for $\rho_\sigma^{(1)}$ using $c_i^{(1)}$, and then we will use $\rho_\sigma^{(1)}$ to solve for $c_i^{(2)}$. It is possible but not necessary for our purposes to iterate further. The kinetic equation for the biomass density $\rho_\sigma$ is,

$$
\dot{\rho}_\sigma = \rho_\sigma \sum_i \alpha_{\sigma,i} \frac{c_i}{K + c_i}.
\tag{22}
$$

Substituting the $\phi$ expansion gives, to leading order,

$$
\begin{aligned}
\dot{\rho}_\sigma^{(1)} &= \rho_\sigma(0) \sum_i \frac{\alpha_{\sigma,i} c_i^{(1)}}{K} \\
&= \rho_\sigma(0) \sum_i \frac{\alpha_{\sigma,i} c_i^{(1)}(0) e^{-\gamma_i t}}{K} \\
\rho_\sigma^{(1)} &= \rho_\sigma(0) \sum_i \frac{\alpha_{\sigma,i} c_i^{(1)}(0)}{K \gamma_i} (1 - e^{-\gamma_i t}).
\end{aligned}
\tag{23}
$$

Taking the long-time limit, $t \gg \frac{1}{\gamma_i}$, we obtain,

$$\rho_\sigma^{(1)}(t \gg \gamma_i^{-1}) = \rho_\sigma(0) \sum_i \frac{\alpha_{\sigma,i} c_i^{(1)}(0)}{K \gamma_i}. \tag{24}$$

Focusing on the leading order, we conclude that,

$$\rho_\sigma(t \gg \gamma_i^{-1}) = \rho_\sigma(0) + \phi \rho_\sigma(0) \sum_i \frac{\alpha_{\sigma,i} c_i^{(1)}(0)}{K \gamma_i}. \tag{25}$$

Substituting for $\gamma_i$ and for $c_i^{(1)}(0) = c_i(0)/\phi$, we have,

$$\rho_\sigma(t \gg \gamma_i^{-1}) \approx \rho_\sigma(0) + \rho_\sigma(0) \sum_i \frac{\alpha_{\sigma,i} c_i(0)}{\sum_{\sigma'} \alpha_{\sigma',i} \rho_{\sigma'}(0)}. \tag{26}$$

Explicitly stating the batch number $d$, at the end of the batch, that is, at time $t = t_f \gg \gamma_i^{-1}$, the biomass density is,

$$\rho_\sigma(d, t_f) \approx \left(1 + \sum_i \frac{\alpha_{\sigma,i} c_i(0)}{\sum_{\sigma'} \alpha_{\sigma',i} \rho_{\sigma'}(d, 0)}\right) \rho_\sigma(d, 0). \tag{27}$$

In the serial dilution model with complete consumption of all nutrients $c_0$ and initial biomass $\rho_0$, the inoculum populations in batch $d + 1$ can be computed from the populations at the time of complete nutrient consumption, $t_f$, in batch $d$,

$$\begin{aligned}
\rho_\sigma(d+1, 0) &= \frac{\rho_0}{\rho_0 + c_0} \rho_\sigma(d, t_f) = \frac{\rho_\sigma(d, t_f)}{1 + c_0/\rho_0} \\
&= \frac{\rho_\sigma(d, 0)}{1 + c_0/\rho_0} \left(1 + \sum_i \frac{\alpha_{\sigma,i} c_i(0)}{\sum_{\sigma'} \alpha_{\sigma',i} \rho_{\sigma'}(d, 0)}\right).
\end{aligned} \tag{28}$$

At steady state, we require that $\rho_\sigma(d+1, 0) = \rho_\sigma(d, 0)$:

$$1 + c_0/\rho_0 = 1 + \sum_i \frac{\alpha_{\sigma,i} c_i(0)}{\sum_{\sigma'} \alpha_{\sigma',i} \rho_{\sigma'}(d, 0)}. \tag{29}$$

Our calculation, to order $\phi$, gives,

$$\frac{c_0}{\rho_0} = \sum_i \frac{\alpha_{\sigma,i} c_i(0)}{\sum_{\sigma'} \alpha_{\sigma',i} \rho_{\sigma'}(0)}. \tag{30}$$

Dividing both sides by $t_f$ and defining,

$$\tilde{\delta} = \frac{c_0}{\rho_0 t_f}, \quad s_i = \frac{c_i(0)}{t_f}, \tag{31}$$

we finally reach the $c_0/K \ll 1$ steady-state condition for the serial dilution system:

$$\tilde{\delta} = \sum_i \frac{\alpha_{\sigma,i} s_i}{\sum_{\sigma'} \alpha_{\sigma',i} \rho_{\sigma'}(0)}. \tag{32}$$

Averaged over a batch, $s_i$ is the average rate that nutrient $i$ is supplied, and $\tilde{\delta}$ is the average rate that all the nutrients are supplied per unit inoculum biomass. If this were a chemostat rather than a serial dilution model, then one could think of $s_i$ as the rate nutrient $i$ is continuously supplied. Moreover, for a chemostat, the parameter $\tilde{\delta}$, which would be the rate all nutrients are continuously supplied per unit biomass, would need to equal $\delta$, the dilution rate of the chemostat to maintain steady state. Indeed, *Equation 32* is precisely the steady-state condition for the chemostat (Equation 4 from *Posfai et al., 2017*) with $s_i$ and $\tilde{\delta} = \delta$ interpreted as above.

Thus we complete the proof that in $c_0/K \ll 1$, the steady state of our serial dilution model is identical to the steady state of the equivalent chemostat model.

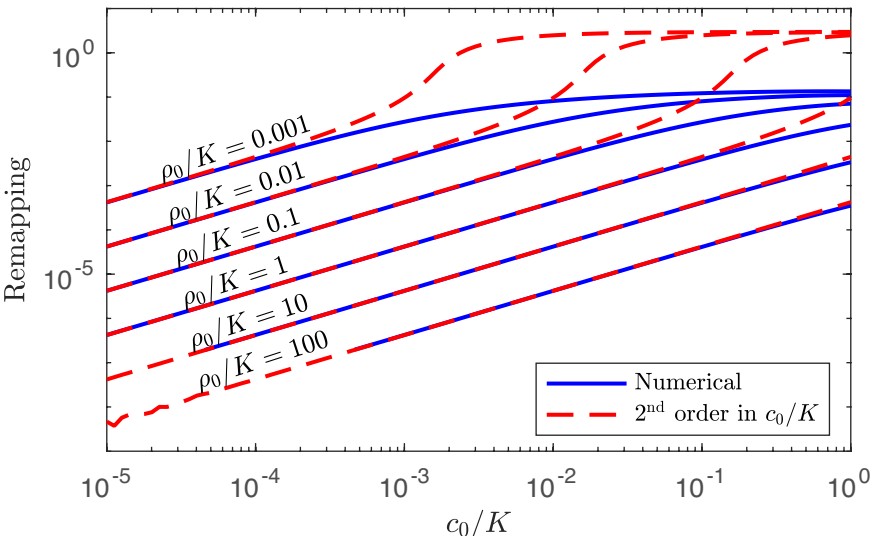

**Appendix 4—figure 1.** Numerical solution and analytical perturbation theory results for remapping of the coexistence boundaries at low $c_0$. The analytic solution is derived from $\forall i : I_i = \mathrm{const}$ using the second-order expansion in *Equation 40*.

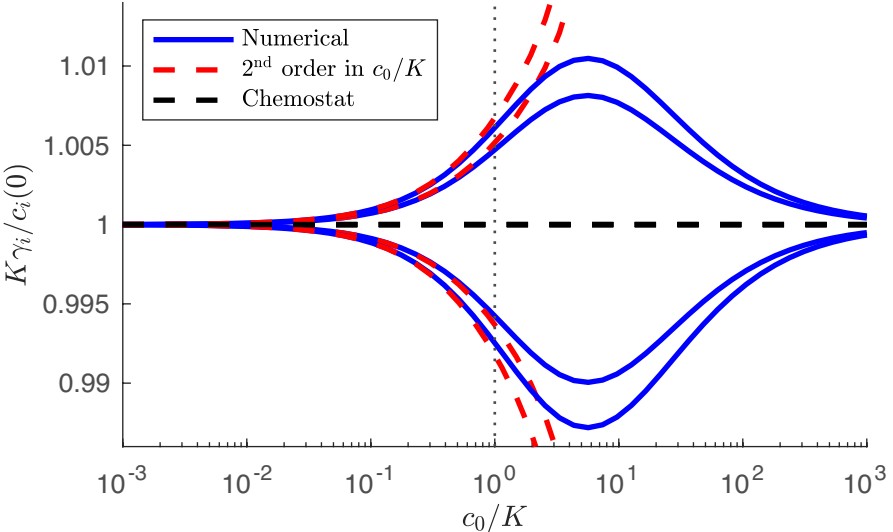

**Appendix 4—figure 2.** Numerical solution and analytical perturbation theory results for the steady-state solution manifold at low $c_0$. The analytic solution is derived from *Equation 47* and the chemostat solution is from *Equation 43*. *Outer curves*: 3 species with strategies $\{(0.1, 0.9), (0.45, 0.55), (1, 0)\}$. *Inner curves*: 3 species with strategies $\{(0, 1), (0.5, 0.5), (1, 0)\}$. In both cases, $\rho_0/K = 1$ and the nutrient supply is $(0.55, 0.45)$.

## Second-order corrections to remapping of the coexistence boundaries for $c_0/K \ll 1$

We have demonstrated above that the leading terms in an expansion for small nutrient supply retrieve the steady-state solution of the chemostat model. However, we know from numerical simulations, that as $c_0/K \equiv \phi$ is increased, the coexistence boundaries become *remapped*, away from the enzyme strategies. Since there is no remapping in the chemostat limit, equivalent to an expansion to

order $\phi$ as proved above, to capture the remapping we expand to order $\phi^2$ in $c_i(t)$. To this end, we return to the $\phi$ expansion and extract the $\phi^2$ contribution,

$$\frac{\dot{c}_i^{(2)}}{K} = \left(\frac{c_i^{(1)}}{K}\right)^2 \gamma_i - \left(\frac{c_i^{(2)}}{K}\right)\gamma_i - \left(\frac{c_i^{(1)}}{K}\right)\sum_\sigma \frac{\alpha_{\sigma,i}\rho_\sigma^{(1)}}{K}. \tag{33}$$

which gives,

$$\frac{c_i^{(2)}}{K} = \left(\frac{c_i^{(1)}(0)}{K}\right)^2 \left(e^{-\gamma_i t} - e^{-2\gamma_i t}\right) - \frac{c_i^{(1)}(0)}{K}\sum_\sigma \frac{\alpha_{\sigma,i}\rho_\sigma(0)}{K}\sum_j \frac{\alpha_{\sigma,j}c_j^{(1)}(0)}{K\gamma_j}\left(te^{-\gamma_i t} - \frac{e^{-\gamma_i t} - e^{-(\gamma_i+\gamma_j)t}}{\gamma_j}\right). \tag{34}$$

Now we can solve for the growth-function integrals $I_i$ for the case of a single species growing in isolation. We expand the growth-function integral,

$$I_i = \int_0^\infty \frac{c_i(t')}{c_i(t') + K}dt' = \phi I_i^{(1)} + \phi^2 I_i^{(2)} + ... \tag{35}$$

Substituting the order $\phi$ from *Equation 22* and integrating, gives:

$$I_i^{(1)} = \int_0^\infty \frac{c_i^{(1)}(t')}{K}dt' = \frac{c_i^{(1)}(0)}{K\gamma_i}. \tag{36}$$

For a single species growing in isolation, $\gamma_i = \frac{1}{K}\alpha_{\sigma,i}\rho_\sigma(0)$. Thus, to order $\phi$, the chemostat limit, we obtain, (in the chemostat limit, for a single species),

$$I_i = \phi I_i^{(1)} = \frac{c_i(0)}{\alpha_{\sigma,i}\rho_\sigma(0)}. \tag{37}$$

Thus, to satisfy the coexistence boundary conditions: $\forall i : I_i = \text{const}$, to order $\phi$ it must be that $\forall i : c_i(0)/\alpha_{\sigma,i} = \text{const}$. This is precisely the coexistence condition for the chemostat, explained in Appendix 3. To obtain the *remapping* of the coexistence boundaries, we must expand $I_i$ to order $\phi^2$.

To order $\phi^2$ we substitute *Equation 22* for $c_i^{(1)}$ and *Equation 34* for $c_i^{(2)}$ and integrate, giving:

$$I_i^{(2)} = \int_0^\infty \left[\frac{c_i^{(2)}}{K} - \left(\frac{c_i^{(1)}}{K}\right)^2\right]dt' = -\frac{c_i^{(1)}(0)}{K}\sum_\sigma \frac{\alpha_{\sigma,i}\rho_\sigma(0)}{K}\sum_j \frac{\alpha_{\sigma,j}c_j^{(1)}(0)}{\gamma_j K}\left(\frac{1}{\gamma_i^2} - \frac{1}{\gamma_i\gamma_j} + \frac{1}{\gamma_j(\gamma_i+\gamma_j)}\right), \tag{38}$$

which upon substituting $\gamma_i$ for a single species simplifies to:

$$\phi^2 I_i^{(2)} = -\frac{c_i(0)}{\rho_\sigma(0)\alpha_{\sigma,i}}\sum_j \frac{c_j(0)}{\rho_\sigma(0)\alpha_{\sigma,j}}\frac{\alpha_{\sigma,j}^2}{\alpha_{\sigma,i}+\alpha_{\sigma,j}}$$

$$= -\phi I_i^{(1)}\sum_j \phi I_j^{(1)}\frac{\alpha_{\sigma,j}^2}{\alpha_{\sigma,i}+\alpha_{\sigma,j}}. \tag{39}$$

Collecting terms to order $\phi^2$ gives

$$I_i = \phi I_i^{(1)}\left(1 - \sum_j \phi I_j^{(1)}\frac{\alpha_{\sigma,j}^2}{\alpha_{\sigma,i}+\alpha_{\sigma,j}}\right) + O(\phi^3). \tag{40}$$

As before, the coexistence boundaries are defined by $I_i = \text{const}$. To order $\phi^2$, *Equation 40* can be used to solve for this remapping analytically. *Equation 40* also clarifies why perfect generalists ($\forall i, j : \alpha_{\sigma,i} = \alpha_{\sigma,j}$) do not get remapped, as stated in the main text. This is because for a generalist $\forall i, j : \alpha_{\sigma,j}^2/(\alpha_{\sigma,i} + \alpha_{\sigma,j}) = \text{const.}$, meaning that the second-order $I_i$ will all be equal if the first-order $I_i$ are all the same. Moreover, the term $I_j^{(1)} \propto \rho_\sigma^{-1}(0)$ multiplies the order $\phi^2$ correction to the coexistence boundary, and as a result, the larger $\rho_\sigma(0)$, the smaller the remapping. A comparison between the analytic form of the remapping at small $c_0/K$ and its numerical form is shown in *Appendix 4—figure*

**1**. As is apparent, the agreement is excellent and extends to higher $c_0/K$ as $\rho_0/K$ increases because at high $\rho_0/K$ the remapping is small.

## Second-order corrections to the steady-state abundance manifold for $c_0/K \ll 1$

Intuitively, one expects that at steady state, the capacity to consume a nutrient will match the total amount of nutrient supplied. Indeed, this was previously shown in the chemostat, and we will extend this statement beyond the chemostat limit, to second order in $c_0/K$. First, we review the results for a chemostat with dilution rate $\delta$ and nutrient supply rate $s_i$ (**Posfai et al., 2017**),

$$\delta = \sum_i \frac{\alpha_{\sigma,i} s_i}{\sum_{\sigma'} \alpha_{\sigma',i} \rho_{\sigma'}^*} \,. \tag{41}$$

We use the asterisk in $\rho_{\sigma}^*$ to make explicit that the abundances are the steady-state abundances, after the system has moved from its initial conditions. As a result, the steady state constrains the abundances $\rho_{\sigma}^*$, such that they must lie on a manifold of solutions that satisfy $\sum_{\sigma} \alpha_{\sigma,i} \rho_{\sigma}^* = \frac{E}{\delta} s_i$. This is precisely the requirement that the total nutrient consumption rate matches the nutrient supply rate.

As demonstrated earlier in this section, by identifying $\tilde{\delta} = \frac{c_0}{\rho_0 t_f}$ and $s_i = \frac{c_i(0)}{t_f}$, to first order in $\phi \equiv c_0/K \ll 1$, the steady-state condition for the serial dilution system is identical to the steady state of the equivalent chemostat. Thus, to order $\phi$,

$$\frac{c_0}{\rho_0} = \sum_i \frac{\alpha_{\sigma,i} c_i(0)}{K \gamma_i} \ (\text{order } \phi). \tag{42}$$

In the serial-dilution framework, to leading order in $\phi$, the chemostat limit of the serial-dilutions steady state is,

$$K \gamma_i = \sum_{\sigma} \alpha_{\sigma,i} \rho_{\sigma}^*(0) = \rho_0 E \frac{c_i(0)}{c_0} + O(\phi^2) \,. \tag{43}$$

We note that $K\gamma_i$ from **Equation 43** is a solution of **Equation 42**, with $\sum_i \alpha_{\sigma,i} = E$. Having established **Equation 43** as the leading order (chemostat limit) term in an expansion in $\phi$, we proceed to calculate $K\gamma_i$ to order $\phi^2$ to obtain corrections to the chemostat limit.

From **Equation 3**, we obtain $\rho_\sigma(t) = \rho_0(0) e^{\sum_i \alpha_{\sigma,i} I_i}$, so that,

$$
\begin{aligned}
I_i &= \phi I_i^{(1)} + \phi^2 I_i^{(2)}, \\
\rho_\sigma(t) = \rho_\sigma(0) &\left[ 1 + \phi \sum_i \alpha_{\sigma,i} I_i^{(1)} + \phi^2 \sum_i \alpha_{\sigma,i} I_i^{(2)} \right. \\
&\left. + \frac{1}{2} \phi^2 \left( \sum_i \alpha_{\sigma,i} I_i^{(1)} \right)^2 \right] + O(\phi^3) \,.
\end{aligned}
\tag{44}
$$

At steady state, dilution at the end of the batch brings the system to its initial conditions, such that for some time $t_f$ when the nutrients in a batch have been consumed, the steady-state species abundances $\rho_\sigma^*(0)$ obey

$$
\begin{aligned}
\rho_\sigma^*(0) &= \frac{\rho_0}{\rho_0 + c_0} \rho_\sigma^*(t_f) \\
&= \frac{\rho_\sigma^*(0)}{1 + c_0/\rho_0} \left[ 1 + \phi \sum_i \alpha_{\sigma,i} I_i^{(1)} + \phi^2 \sum_i \alpha_{\sigma,i} I_i^{(2)} \right. \\
&\left. + \frac{1}{2} \phi^2 \left( \sum_i \alpha_{\sigma,i} I_i^{(1)} \right)^2 \right] \,.
\end{aligned}
\tag{45}
$$

Indeed, $\rho_\sigma^*(0)$ cancels, yielding the serial-dilutions steady-state to order $\phi^2$,

$$\frac{c_0}{\rho_0} = \phi \sum_i \alpha_{\sigma,i} \tilde{I}_i^{(1)} \left[ 1 + \frac{\phi}{2} \sum_j \alpha_{\sigma,j} \tilde{I}_j^{(1)} + \phi \frac{\tilde{I}_i^{(2)}}{\tilde{I}_i^{(1)}} \right]. \tag{46}$$

We have added a tilde sign, as in $\tilde{I}_i^{(1)}$, to stress that we use leading order in $\phi$, having already explicitly accounted for $\phi$ in the expansion, and so, take $\phi \tilde{I}_i^{(1)} = \frac{c_0}{\rho_0 E}$. Substituting for $\tilde{I}_i^{(2)}$ similarly, reduces *Equation 46* to,

$$
\begin{aligned}
K\gamma_i &= \sum_\sigma \alpha_{\sigma,i} \rho_\sigma^*(0) \\
&= c_i(0) E \frac{\rho_0}{c_0} \left[ 1 + \frac{c_0}{\rho_0} \left( \frac{1}{2} - X_i \right) \right], \\
X_i &= \frac{c_0}{\rho_0 E^2} \sum_\sigma \alpha_{\sigma,i} \rho_\sigma^*(0) \sum_j \alpha_{\sigma,j} \frac{c_j(0)/c_i(0)}{c_i(0) + c_j(0)}.
\end{aligned}
\tag{47}
$$

Comparing *Equation 47* with *Equation 43* we note the small, order $c_0/\rho_0$, corrections to the chemostat limit. We overlay the perturbation theory solution, *Equation 47*, on the numerical solution, showing good agreement for $c_0/K < 1$, plotted in *Appendix 4—figure 2*. We note that for the case of balanced nutrients, $\forall i : c_i(0) = c_0/p$, we have $X_i = \frac{1}{2}$, and therefore, when the nutrient supply is balanced, there are no second order corrections to the chemostat solution. Moreover, for balanced supply, the exact numerical solution also does not deviate from the chemostat limit.

## Appendix 5

### Remapping of the coexistence boundaries for $c_0/K \gg 1$

Here we show that at high $c_0/K$ the coexistence boundaries remap to their chemostat positions. When a large nutrient bolus is present, the growth function is effectively always saturated such that

$$\frac{d\rho_\sigma}{dt} = \rho_\sigma \sum_{i=1}^{p} \alpha_{\sigma,i} \frac{c_i}{K+c_i} \approx \rho_\sigma \sum_i \alpha_{\sigma,i} = \rho_\sigma E, \tag{48}$$

where $\sum_i \alpha_{\sigma,i} = E$ is in units of 1/time, without loss of generality $E$ can be set to unity, but we keep it here to make the units explicit. Solving for $\rho(t)$ yields $\rho(t) = \rho_0 e^{Et}$. The assumption that $c_i \gg K$ can then be applied to the nutrient dynamics, yielding:

$$\frac{dc_i}{dt} = -\alpha_{\sigma,i}\rho_\sigma(t)\frac{c_i}{K+c_i} \approx -\alpha_{\sigma,i}\rho_0 e^{Et}. \tag{49}$$

Solving for the nutrient dynamics leads to $c_i(t) = c_i(0) + \frac{\alpha_{\sigma,i}}{E}\rho_0(1 - e^{Et})$. Since the growth function is nearly always saturated (giving an integrand value of 1), the growth-function integral $I_i = \int_0^\infty \frac{c_i}{K_i+c_i}\,dt$ approximately equals the time of nutrient exhaustion. Thus for a given nutrient $i$, the time when that nutrient is depleted $t_{i,f}$ is given by:

$$t_{i,f} = I_i = \frac{1}{E}\ln\left(1 + \frac{c_i(0)E}{\alpha_{\sigma,i}\rho_0}\right). \tag{50}$$

Note that the coexistence boundaries are defined by $\forall i : I_i = \text{const}$ which is satisfied when the fraction of nutrients in the initial bolus matches the strategies,

$$\forall i : c_i(0)/\alpha_{\sigma i} = \text{const}. \tag{51}$$

This is precisely the result in Appendix 3, indicating that in the $c_0/K \gg 1$ limit the coexistence boundaries return to their chemostat values.

## Appendix 6

### Comparison of growth model assumptions with experimental data

We have explored the implications of our model in a variety of contexts, but our modeling framework drastically simplifies bacterial growth, ignoring many factors relevant for microbial coexistence, e.g., lag times for the recovery of growth and various responses to stresses including starvation. In this section we compare certain key aspects of our growth model assumptions with experimental data.

A well-known form of nutrient utilization in the microbiology literature is sequential utilization, where a preferred sugar (often glucose) is consumed before others (*Monod, 1942*). However, this mechanism applies to high sugar levels (on the order of grams per liter), such as those found in laboratory media. Many natural environments, such as marine systems and feces, contain low concentrations of sugars (*Münster, 1993*; *Flourie et al., 1986*). At such low concentrations, simultaneous utilization of multiple substitutable sugars is observed (*Kovárová-Kovar and Egli, 1998*; *Egli et al., 1993*; *Lendenmann et al., 1996*). We therefore compared our modeling for single-species growth to previously published data from chemostat and batch experiments on *E. coli* supplied with multiple sugars at low concentrations.

We use chemostat data from *Lendenmann et al., 1996* who measured the steady-state concentrations of biomass and sugars, with *E. coli* continuously supplied with mixtures of glucose, fructose, and ribose. We applied the chemostat version of the model (*Posfai et al., 2017*) and constrained the fit with previously measured values of the Monod constants $K_i$ for this strain (*Lendenmann and Egli, 1998*). From the fit, we estimated the consumption strategies $\alpha_i$ for glucose, fructose, and ribose, with the rest of the parameters being defined experimentally (see the end of this section for details of the fitting procedure). As shown in *Appendix 6—figures 1A and B*, the resulting model matches the data well with strategies, measured in $(\text{mg sugar})(\text{mg biomass})^{-1}\text{h}^{-1}$, of $\alpha_{gluc} = 1.96 \pm 0.12$, $\alpha_{fruc} = 2.04 \pm 0.11$, and $\alpha_{ribo} = 1.41 \pm 0.01$. This corresponds to a normalized strategy of (0.36, 0.38, 0.26). The only notable deviations between the best-fit model and the data occurs for two fructose steady states. These deviations would be corrected if $K_{\text{fruc}}$ was larger, suggesting that the $K_{\text{fruc}}$ used here may not reflect the actual value in the experiment. The model also accurately predicts the resulting steady-state biomass concentrations, which are a constant 47 mg/L in the experiment and approximately constant at 45 mg/L in our model. This agreement suggests that our growth model assumptions are consistent with the behavior of *E. coli* growing at low nutrient concentration with a continuous nutrient supply. Despite being supplied with a variety of different sugar mixtures, *E. coli* maintains a constant steady-state biomass in these experiments because all of the carbon sources are substitutable.

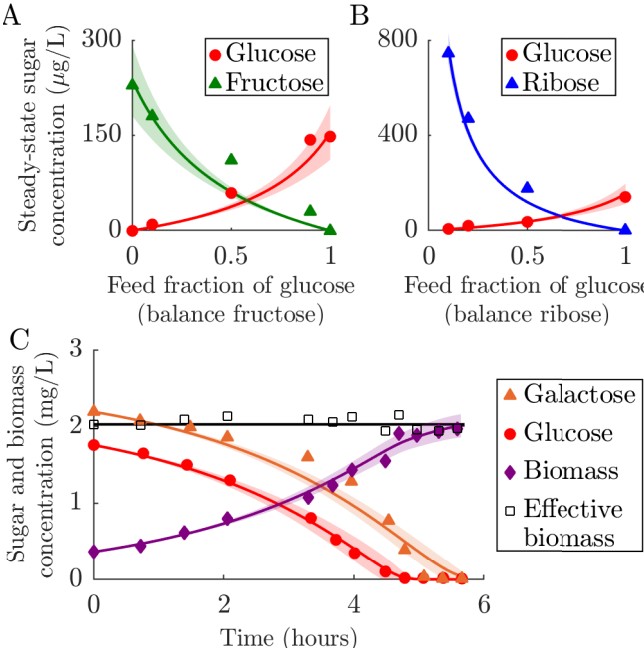

**Appendix 6—figure 1.** Fitting of fixed-enzyme-budget model to experimental data. (**A–B**) Fit of the chemostat version of the model to data from chemostat experiments from *Lendenmann et al., 1996*. The experimental data are steady-state concentrations of sugars in *E. coli* chemostats supplied with different mixtures of glucose, fructose, and ribose. The strategy $\alpha_i$ for each sugar is inferred, whereas all other parameters are derived from the experimental conditions and measurements. The solid curves show the model prediction, with the shaded region marking the 95% prediction bound (see Appendix 1 for details). (**A**) Comparison of model to data from chemostats supplied with glucose and fructose with a constant total feed concentration of 100 mg/L. (**B**) Comparison of model to data from chemostats supplied with glucose and ribose with a constant total feed concentration of 100 mg/L. (**C**) Comparison of serial dilution model fit to batch growth data from *Egli et al., 1993*. Solid curves are model predictions and the shaded area is the 95% prediction bound. 'Effective biomass' refers to the total biomass within the system: $M(t) = \rho(t) + Y(c_{gluc}(t) + c_{gal}(t))$. Since the data for the three timeseries were measured at slightly different times, the effective biomass for the experimental data was obtained by linear interpolation of the data points. The inferred parameters were the strategy $(\alpha_1, \alpha_2)$ for the two sugars, glucose and galactose, and the yield $Y$.

To explicitly test growth dynamics, though for a single species only, we compared our model to batch growth data from *Egli et al., 1993*. In this experiment, timecourses of biomass and nutrient concentrations were measured in a culture of *E. coli* supplied with a mixture of glucose and galactose. The *E. coli* used to seed this culture came from a glucose-limited chemostat (we also compared our model to a batch seeded with *E. coli* from a galactose-limited chemostat, see *Appendix 6—figure 2*). For this data we used our serial dilution model with Monod kinetics and $K_i$ values from measurements on the same strain (*Lendenmann and Egli, 1998*). As shown in *Appendix 6—figure 1C*, the agreement between the best-fit model and the experimental data is generally quite good over the entire time course. The estimated $\alpha_i$, measured in units of (mg sugar)(mg biomass)$^{-1}$h$^{-1}$, were $0.46 \pm 0.04$ and $0.41 \pm 0.03$ for glucose and galactose, respectively. The estimated yield was $0.42 \pm 0.03$, similar to the experimentally measured yield used in the chemostat model of 0.45 (mg biomass)(ms sugar)$^{-1}$. Our model captures the glucose and biomass trends very well, but some galactose data points fall outside of the confidence interval. In addition to possible experimental noise, this may be due to small variations in yield during growth, as a constant yield would imply that accurately modeling biomass and glucose would necessarily also accurately capture galactose ($c_{gal}(t) = const - \rho(t) - Yc_{gluc}(t)$). Qualitatively, the data support the assumptions of substitutable and simultaneous nutrient consumption, while the strong quantitative fit to our model supports the assumption that enzyme strategies do not vary significantly within a batch.

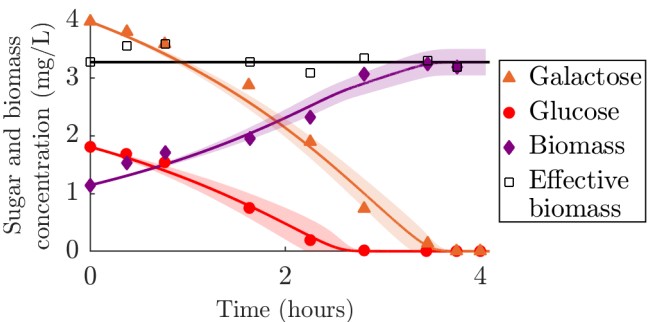

**Appendix 6—figure 2.** Comparison of serial dilution model fit to batch growth data from *Lendenmann et al., 2000*. This data is similar to that in *Appendix 6—figure 1C*, except that the inoculum was taken from galactose-limited conditions instead of glucose limited conditions. Solid curves are model predictions and the shaded area is the 95% prediction bound (see Appendix 1 for details). 'Effective biomass' refers to the total biomass within the system: $M(t) = \rho(t) + Y(c_1(t) + c_2(t))$. Since the data for the three timeseries were measured at slightly different times, the effective for the experimental data was obtained by linear interpolation of the data points. The inferred parameters were the strategy ($\alpha_1$, $\alpha_2$) for the two sugars, glucose and galactose, and the yield $Y$. The estimated $\alpha_i$, measured in units of (mg sugar)(mg biomass)$^{-1}$h$^{-1}$, were $0.43 \pm 0.06$ and $0.57 \pm 0.04$ for glucose and galactose, respectively. The estimated yield of $0.37 \pm 0.03$ was similar to that inferred in *Appendix 6—figure 1C*.

While we only compare our model to data from *E. coli*, substitutable and simultaneous growth on multiple nutrients has been observed in other bacteria such as *Lactobacillus brevis* (*Kim et al., 2009*), and has even been observed in non-prokaryotic organisms. For example, the eukaryote *Kloeckera sp. 2201* has been shown to simultaneously utilize methanol and glucose as carbon sources (*Kovárová-Kovar and Egli, 1998*). Similarly, the methanogenic archaeon *Methanosarcina barkeri* can simultaneously utilize methanol and acetate in batch culture (*Scherer and Sahm, 1981*). However, it should be noted that our simple model cannot describe the growth kinetics of all microbes in all conditions. For example, the inferred strategy of *E. coli* for glucose varied between the batch and chemostat experiments examined here, suggesting that the total metabolic enzyme budget of microbes changes in different conditions. Such variation is likely due to other cell functions, such as ribosome synthesis (*Scott et al., 2010*), consuming different fractions of the cell's total material and energy budget, something we do not explicitly model. Indeed, our goal is not to precisely model all microbial growth phenomenon, but rather to construct a widely applicable approximation of microbial growth in order to better understand ecological dynamics.

## Fitting to experimental data

To fit our model to the experimental data in *Lendenmann et al., 1996*, we first digitally extracted the steady-state data points from the experimental figures. We used the model from *Posfai et al., 2017* with Monod kinetics. The $K_i$ of glucose, ribose, and fructose were taken as 73, 132, and 125 µg/L sugar, respectively (*Lendenmann and Egli, 1998*). The model was fit to the data and the standard error of parameters were estimated using the MATLAB curve fitting toolbox. The only parameters estimated were the $\alpha_i$ of glucose, ribose, and fructose. It was assumed that all sugars had a biomass yield of $Y = 0.45$, as measured experimentally (*Lendenmann et al., 1996*). The supply rates for a given simulation were computed as $S_i = c_{f,i}\delta$, where $S_i$ is the nutrient supply rate of nutrient $i$, $c_{f,i}$ is the concentration in the feed of nutrient $i$, and δ is the dilution rate of the chemostat. The fitting process minimized only the sum of squared errors between the model and the nutrient concentration data, since steady-state biomass within the model is approximately constant and determined by measured parameters ($\rho_{ss} \approx \frac{Y\sum_i S_i}{\delta}$). Confidence intervals for parameters were estimated using MATLAB's *confint* function, which computes the interval using an estimate of the diagonal elements of the covariance matrix of the coefficients multiplied by the inverse of the Student's *t* distribution. The prediction bounds (shaded regions) are calculated using MATLAB's *predint* function, which uses

the estimated covariance matrix and the Jacobian of the fitted values to the parameters to predict the bounds.

The data fitting procedure for the batch experimental data was similar to that employed for the chemostat experimental data. We digitally extracted the data points from the figure in *Egli et al., 1993* and used the MATLAB curve fitting toolbox. The biomass was reported as $OD_{546}$ and was converted to mg/L using a conversion factor measured for the same strain (*Lendenmann et al., 1996*). Two sugar data points that were taken before the first biomass measurement were removed so that the initial conditions of the system would be well-defined. We estimated the parameters of the serial dilution model developed in this paper assuming Monod kinetics (*Equation 1*). It was further assumed that both sugars had the same yield, *Y*. The yield was not measured in the experimental study and was therefore left as a fitting parameter. The $K_i$ for glucose and galactose were 73 and 98 µg/L, respectively (*Lendenmann and Egli, 1998*). The three fitting parameters were the yield *Y* and the strategies $\alpha_i$ for glucose and galactose. The data points of the sugar and biomass measurements were taken at slightly different times, so the effective biomass for the experimental data was obtained by linear interpolation of the data points. Confidence intervals and prediction bounds were estimated using the same methods as for the chemostat model.

## Appendix 7

### Supplemental figures

In this section we present supplemental figures that support the main text. Each figure's caption contains all pertinent information.

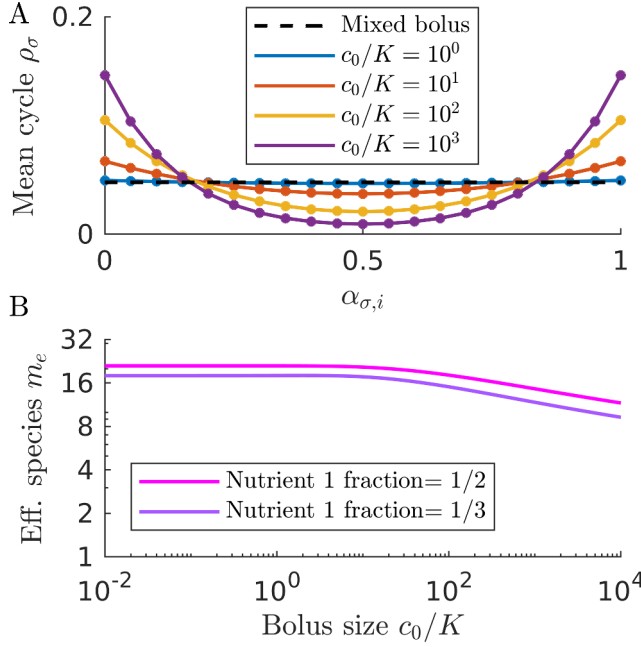

**Appendix 7—figure 1.** Serial dilution model with cycling bolus compositions. In this work we largely consider the case of nutrient boli that contain a defined mixture of nutrient. However, in nature the nutrient boli may not themselves contain a mixture of nutrients, but instead approach a mixed distribution of nutrients over time. To explore we compare the case of mixed boli with that of cycled single-nutrient boli that are varied in time to approach a mixed distribution. (**A**) Mean steady-state population abundances in communities supplied with boli containing an equal mixture of two nutrients (dashed line) or alternating boli each containing a single nutrient (solid curves). Population abundances are averaged over a single cycle. The community is composed of 21 equally-spaced species. For the cycling bolus case, the species that are more specialized for either nutrient become more abundant due to a 'single-nutrient' early-bird effect as $c_0/K$ is increased. (**B**) Effective number of species as a function of $c_0/K$ for communities supplied with cycling single nutrient boli. Communities consist of 21 equally spaced strategies are supplied a cycle of boli approaching an average nutrient 1 fraction of 1/2 or 1/3. The effective number of species decreases as a function of $c_0/K$ due to the 'single-nutrient' early-bird effect.

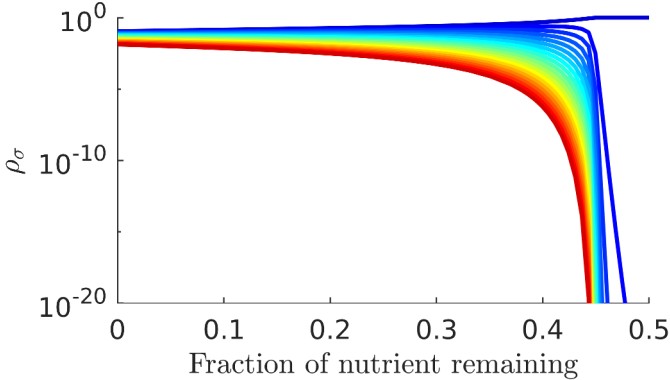

**Appendix 7—figure 2.** Serial dilution model with incomplete nutrient depletion. We have thus far assumed that batches run until the nutrient is entirely depleted. However, batches might be terminated before nutrients are completely depleted. Here we characterize the steady state of a community of 21 equally spaced species when the batch is terminated early such that $\sum c_i/c_0 = \Delta$, where $\Delta$ is the fraction of nutrient remaining at the end of the batch. In these simulations, $\rho_0 = c_0 = K = 1$, with nutrient composition $(1/3, 2/3)$. Batches are repeated until either a relative error tolerance is met (less than $10^{-8}$ change between batches) or 40,000 batches have elapsed (the large batch limit is there to account for possible critical slowing down). As can be seen, coexistence is fairly robust with respect to incomplete nutrient consumption until $\Delta \approx 0.45$ after which point diversity rapidly collapses. The reduction in diversity in the system can be explained by the early-bird effect. In a batch where complete nutrient depletion occurs, the early bird gains an early advantage by rapidly depleting the more abundant nutrient, and then is able to consume a larger share of the non-abundant nutrient. Therefore, if the batch is terminated early, the amount of non-abundant nutrient consumed within the batch becomes smaller. While this makes the early bird less able to consume the non-abundant nutrient, it more severely impacts non-early-bird species, as their growth is more reliant on the non-abundant nutrient. As the batch terminates earlier and earlier, the system effectively becomes competition for a single nutrient (the more abundant one). Thus, the most fit early bird (the specialist for the more abundant nutrient) completely takes over the population.

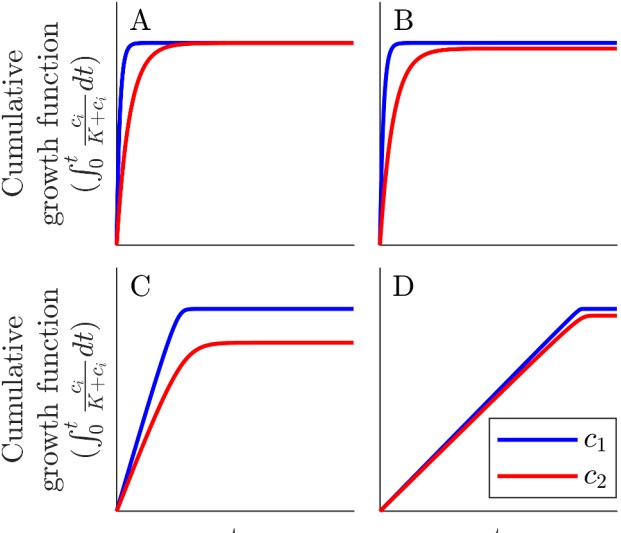

**Appendix 7—figure 3.** Cumulative growth function integrals at different values of $c_0/K$ for a species with $\alpha_\sigma = (0.8, 0.2)$ growing with nutrient supplied with proportion $(0.8, 0.2)$ and initial population $\rho_0 = 1$. (**A**) Cumulative growth integrals with $c_0/K = 0.001$. (**B**) Cumulative growth integrals with $c_0/K = 0.1$. (**C**) Cumulative growth integrals with $c_0/K = 10$. (**D**) Cumulative growth integrals with $c_0/K = 100$. When $c_0 \ll K$ and $c_0 \ll \rho_0$, the consumption rate of each nutrient is proportional to its

own abundance, $c_i/(K + c_i) \approx c_i/K$, and there is little relative change in biomass, $\rho(t) \approx \rho_0$. The more abundant nutrient is consumed faster (since the strategy is matched to nutrient proportions) and the majority of it is consumed quickly. The less abundant nutrient is consumed more slowly and a significant portion of it remains after the more abundant nutrient is almost completely depleted. In this way, the growth timecourse integrals are balanced to be equal. Once $c_0$ increases relative to $\rho_0$, but $c_0$ is not large compared to $K$ this balance is broken. The more abundant nutrient will still be depleted quickly. However, now that $c_0$ is larger this initial consumption results in an increased abundance of the consumer. This means that the less abundant nutrient is depleted more quickly, leading a smaller growth timecourse integral for this nutrient relative to the more abundant nutrient. Thus, in this regime, the difference between the growth timecourse integrals increases with $c_0$. Restoring them to equality requires more equal starting distributions of nutrients. Once $c_0$ increases such that $c_0 \gg K$ and $c_0 \gg \rho_0$, the increase in $c_0$ now drives the growth timecourse integrals towards equality. The growth function is now almost always saturated and this neutralizes the effect of one nutrient starting at a much larger concentration. There will now be a significant buildup of biomass before the nutrients are exhausted, meaning that the growth timecourses will subsequently drop very quickly. As the growth function becomes more and more saturated, the nutrients will be consumed in proportion to their strategy. Thus, the growth timecourse integrals will once again be equal since the strategies match the nutrient proportions and the 'crash' times will therefore be similar for both nutrients.

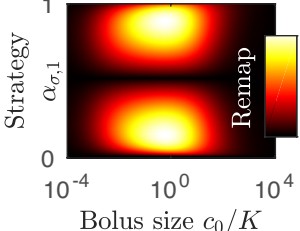

**Appendix 7—figure 4.** Dependence of coexistence boundary remapping on $c_0/K$. As a further exposition to *Figure 3A* in the main text, shown here is the difference between the remapped coexistence boundaries and the corresponding metabolic strategies as a function of $c_0/K$ and metabolic strategy with $\rho_0/K = 10^{-3}$.

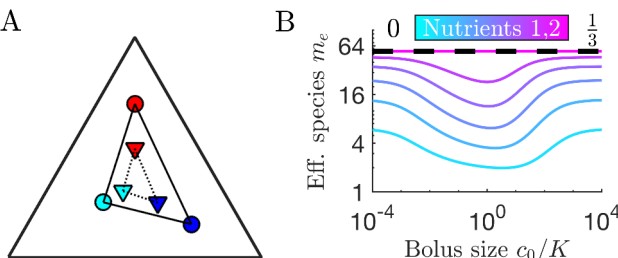

**Appendix 7—figure 5.** Serial dilution model with three nutrients. (**A**) Example of remapping on the three-nutrient simplex, similar to *Figure 2* from the main text. Here we show how the remapping analysis presented in the main text for two nutrients can be extended to three nutrients. Remapping of three strategies for $c_0/K = 1$ and $\rho_0 = 10^{-2}$. Outer circles: strategies $\{\vec{\alpha}_\sigma\}$; inner triangles: remapped nodes; lines connecting outer circles: supplies within this convex hull of strategies lead to coexistence of all species in the chemostat regime $c_0/K \ll 1$; dashes connecting inner circles: approximate remapped convex hull boundary defining region of supplies leading to coexistence for $c_0/K = 1$. Note that, as in the two-nutrient case, the strategies map inwards on the simplex for $c_0/K \approx 1$. (**B**) Steady-state effective number of species $m_e$ versus $c_0/K$ for equal initial inocula of 64 species equally spaced throughout the triangular simplex competing for three nutrients. Effective number of species shows the same trend of loss in diversity when $c_0/K \approx 1$ as in the two-nutrient case in *Figure 2C*.

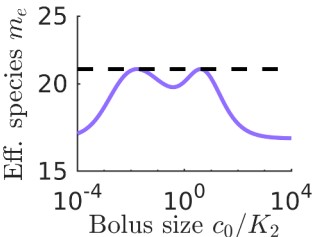

**Appendix 7—figure 6.** Large differences between $K_i$ values can lead to multi-peaked relationships between diversity and bolus size. Here, we present a magnified version of the community growing with nutrient compositon $(0.3, 0.7)$ in *Figure 4A*, with $K_1 = 10^{-3}$ and $K_2 = \rho_0 = 1$. The change in the identity of the early bird can explain how multiple diversity peaks occur in the curve shown. If the system is near maximum diversity in the chemostat limit and the early-bird effect favors the species that is disadvantaged in the chemostat, the system will initially be driven towards maximum diversity with increasing $c_0$. However, as the early-bird effect continues to strengthen, the formerly disadvantaged species will begin to dominate the community, lowering community diversity. Then, as $c_0$ continues to increase and the early-bird effect weakens, the early bird's dominance will wane, again driving the system towards maximum diversity. Finally, the non-early-bird species will overtake the early bird in the high $c_0$ chemostat limit, driving diversity back downwards.

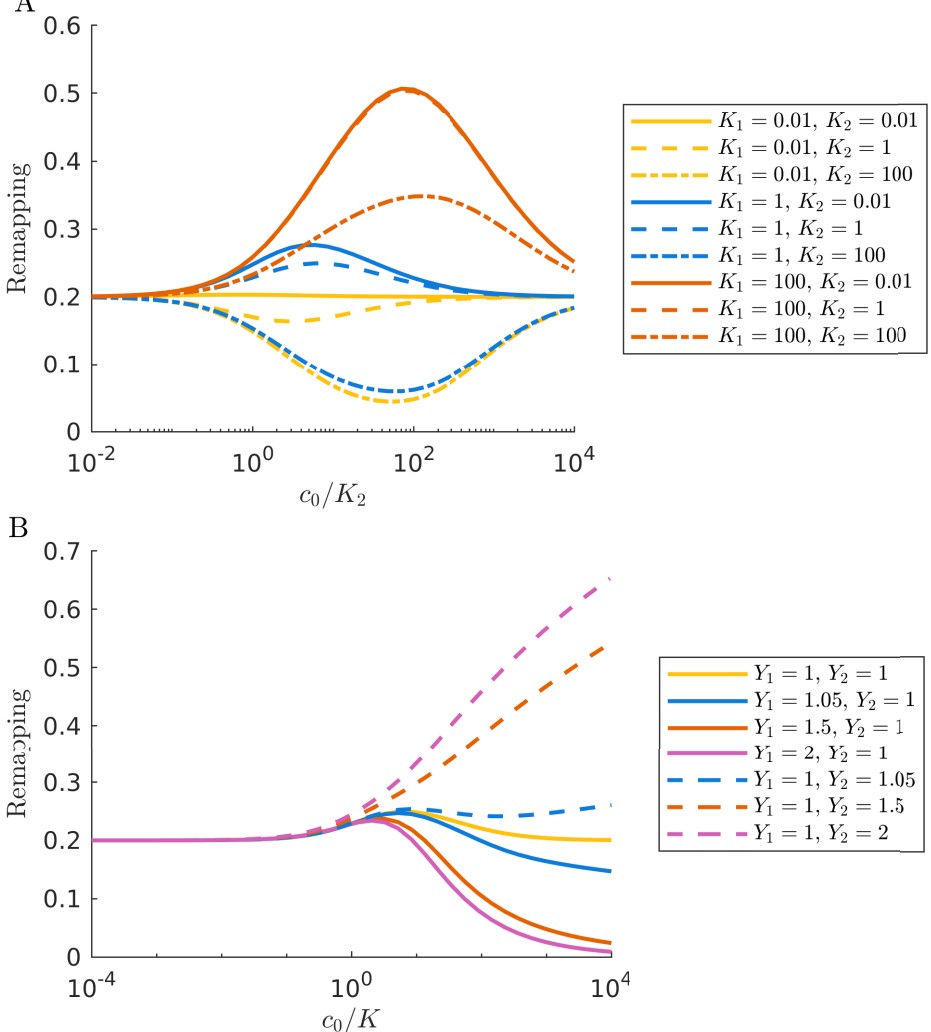

**Appendix 7—figure 7.** Remapping in the serial dilution model with unequal $K_i$ and $Y_i$. Here we show

that varying the values of $K_i$ and $Y_i$ can influence the direction and magnitude of the remapping, using $\alpha_{\sigma 1} = 0.2$ with $\rho_0 = 1$ as an example. (**A**) Remapping with different combinations of $K_i$. The strategy we are examining devotes most of its enzyme budget to consuming nutrient 2. When there is a large different in $K_i$ that favors nutrient 2, inward remapping is enhanced. When there is a large difference that favors nutrient 1, outward remapping is enhanced. Eventually, the remapping returns to the chemostat limit. (**B**): Remapping with different combinations of $Y_i$. Note that when yields are variable the condition for the remapped point is $I_i = \int_0^\infty Y_i \frac{c_i}{K_i + c_i} dt = \text{const}$. The remapped points shown are normalized to yield (if the remapped point is $(x, 1-x)$ the normalized form is $(x^*, 1-x^*)$ where $x^* = \frac{Y_1 x}{Y_1 x + (1-x) Y_2}$). Similar to the unequal $K_i$ case, inward remapping is enhanced when there are large differences in $Y_i$ favoring nutrient 2. When the $Y_i$ favor nutrient 1, outward remapping is enhanced. Unlike in the unequal $Y_i$ case, the remapping does not return to the chemostat limit.

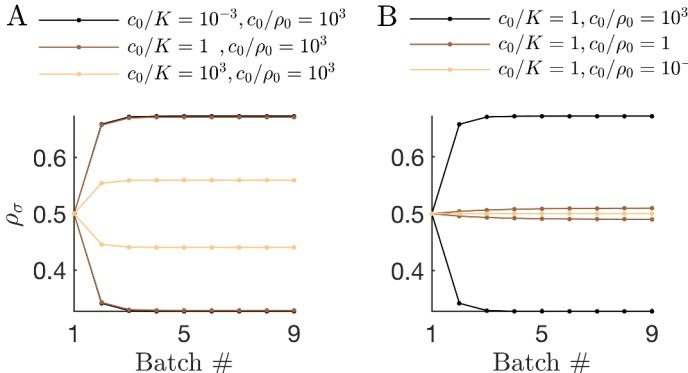

**Appendix 7—figure 8.** Batch timecourses of a bitrophic model with only two species. To further investigate the difference between the unitrophic and bitrophic scenarios, we consider a toy system with only two species, Species 1 with strategy $(0.05, 0.95)$ and Species 2 with strategy $(0.95, 0.05)$. We set the byproduct matrix for perfect conversion, $\Gamma_{1,2} = 1$ and the nutrient bolus composition so that only Nutrient 2 is provided. By the end of each batch, the same amounts of Nutrient 1 and Nutrient 2 have been consumed. (**A**) Simulations with constant $c_0/\rho_0$ and variable $c_0/K$. The 'early-bird' effect becomes stronger with decreasing $c_0/K$, since this allows the dominant species to grow more before the byproduct can be readily consumed. (**B**) Simulations with variable $c_0/\rho_0$ and constant $c_0/K$. The 'early-bird' effect is stronger at higher $c_0/\rho_0$ because the larger amount of supplied nutrient allows the dominant species to build a large population which can then outcompete other species. Conversely, if instead of cross-feeding we were to supply in the nutrient bolus equal quantities of Nutrients 1 and 2, the result would be equal abundance of both species.

