## [Decision Letter]

**Acceptance summary:**

This paper provides an elegant modeling framework for improving our understanding of how nutrients influence the diversity of microbial communities. As such, it will be a valuable addition to the field.

**Decision letter after peer review:**

Thank you for submitting your article "Nutrient levels and trade-offs control diversity in a model seasonal ecosystem" for consideration by *eLife*. Your article has been reviewed by three peer reviewers, and the evaluation has been overseen by a Reviewing Editor and Detlef Weigel as the Senior Editor. The following individual involved in review of your submission has agreed to reveal their identity: Antonella Succurro (Reviewer #1); Lei Dai (Reviewer #3).

The reviewers have discussed the reviews with one another and the Reviewing Editor has drafted this decision to help you prepare a revised submission.

Summary:

Overall, the reviewers liked the work, and they all agreed that this is an insightful paper that should be suitable for publication if the authors address a few issues, none of which requires additional experimentation. Notably, all reviewers thought that the manuscript was a bit difficult to read at times, and they agreed that it would be substantially improved if the authors could shorten the paper and provide a more streamlined narrative, with a focus on its central results. Finally, the reviewers were skeptical of the comparison to experiments and indicated that it should be removed from the Abstract and re-framed as suggested below.

The authors set out to understand the conditions in which n (n > 1) species can coexist on m < n resources. This is important because such coexistence on few resources appears common in nature. Yet, some simple competition models suggest it is impossible (hence the name of the conundrum as "the paradox of the plankton"). They seek to understand the conditions leading to stable diversity in a specific environmental mode: seasonal batch-cultures with bottle-necks, a la serially culturing microbial communities, as they rightfully claim that theory here is less developed than in chemostat.

The most interesting finding in the paper is that nutrient abundance ("bolus") at the onset of each "season" is an important determinant of diversity, whereas reservoir concentrations are less important in a chemostat. The reason is what they call an "early-bird" effect: medium nutrient boluses tend to favor species whose consumption vectors are most similar to the bolus vector, allowing these species to rapidly outpace other species and then pre-empt those other species from resources. In contrast, with lower or much higher nutrient boluses, and especially, when the bolus size is small enough that the model is akin to a chemostat, the fixed energy budget among species causes them to equalize and allows for coexistence. A second finding is that there is a simple mechanism (using their pictorial simplexes) to predict diversity among a set of species, given an understanding of their nutrient consumption vectors and the vector of the nutrient supply. The non-monotonic relation between nutrient bolus size and remapping of strategies, which determines species co-existence at steady state, is a novel and interesting result that can be tested by experiments.

Essential revisions:

1) All reviewers agreed that the two central and most insightful results are (i) the relationship between the bolus size and diversity and the associated remapping of metabolic strategies and (ii) the connection between serial passaging and chemostat modes. The paper would benefit by making it shorter and with a more clear focus on those two results. In particular the second result ((ii) above) could be highlighted in the Abstract / Introduction / Discussion. The authors clearly show that under certain conditions (notably, c_0_ << K), a chemostat model is a special case of a serial dilution model. This is a fascinating result. However, more effort could be taken to detail some of the important assumptions in this result. The main one is that batch culture is allowed to reach steady-state and full nutrient depletion. While this is reasonable from the goal of mathematically connecting the two environmental regimes, it is difficult to consider whether it is ecologically reasonable without some discussion of the amount of time required for that to be reached. For example, are there any batch culture simulations that would require an unreasonably long time to reach steady-state, such that it is unlikely to occur, and therefore unlikely to be equivalent to chemostat steady-state conditions?

2) Many of the other results besides the two highlighted above could be organized in the appendix, or as subsections of a final Results section, where the authors examine how relaxing some of the simplifying assumptions made in the model (e.g. the exploration of a softer trade-off imposed by unequal enzyme budgets, the regulation of enzyme expression, the possibility of cross-feeding, the case of unequal K_i_ and Y_i_, or the effect of including the stochastic nature of dilutions) affect their conclusions. If the authors prefer to include it as a Results section in the main text, they should introduce this section by explicitly alerting the reader that its purpose is to check the effect of relaxing the simplifying assumptions they previously made on the main results of the paper, listing them before going into details. That will help the reader understand the connection of the various subsections to the main results.

3) Reviewers appreciated the Mutation/Selection section, but they all agreed that it does not add much to the narrative. Cutting it off the paper would help to maintain the focus and streamline the narrative. The authors could develop it further in a separate paper, as it is an interesting and relevant topic

4) The comparison to experimental results should be revised. The analysis performed (only one species, not an ecosystem) in Figure 7 does not work as much as a "validation" of the main predictions of the paper, but rather as "support of the model assumptions". The authors described this as: "Qualitatively, the data support the assumptions of substitutable and simultaneous nutrient consumption, while the strong quantitative fit to our model supports the assumption that enzyme strategies do not vary significantly within a batch." The reviewers concurred that validation experiments of the model in this paper would require testing the predictions of remapping: e.g. how co-existing species change as a function of nutrient bolus size. The paper is valid and interesting as it is and without an experimental validation of this prediction, so the current discussion on experiments should be either eliminated or moved to the appendix and changed in tone to reflect its value as a check of some of the model assumptions.

---

## [Author Response]

Summary:Overall, the reviewers liked the work, and they all agreed that this is an insightful paper that should be suitable for publication if the authors address a few issues, none of which requires additional experimentation. Notably, all reviewers thought that the manuscript was a bit difficult to read at times, and they agreed that it would be substantially improved if the authors could shorten the paper and provide a more streamlined narrative, with a focus on its central results. Finally, the reviewers were skeptical of the comparison to experiments and indicated that it should be removed from the Abstract and re-framed as suggested below. […]

We have followed the reviewers’ suggestions and have shorted and streamlined the narrative. As requested, we have reframed the comparison to experiments, moving it to the Appendix (details below).

Essential revisions:1) All reviewers agreed that the two central and most insightful results are (i) the relationship between the bolus size and diversity and the associated remapping of metabolic strategies and (ii) the connection between serial passaging and chemostat modes. The paper would benefit by making it shorter and with a more clear focus on those two results. In particular the second result ((ii) above) could be highlighted in the Abstract / Introduction / Discussion. The authors clearly show that under certain conditions (notably, c_0_ << K), a chemostat model is a special case of a serial dilution model. This is a fascinating result. However, more effort could be taken to detail some of the important assumptions in this result. The main one is that batch culture is allowed to reach steady-state and full nutrient depletion. While this is reasonable from the goal of mathematically connecting the two environmental regimes, it is difficult to consider whether it is ecologically reasonable without some discussion of the amount of time required for that to be reached. For example, are there any batch culture simulations that would require an unreasonably long time to reach steady-state, such that it is unlikely to occur, and therefore unlikely to be equivalent to chemostat steady-state conditions?

We agree with the reviewers that the connection between serial passaging and chemostat models is a central result. As the reviewers suggest, we now dedicate a short section specifically to this connection. We now also highlight the connection to the chemostat in the Abstract and Discussion.

There are certain regimes in our model that do require a large number of batches to reach steady-state. This occurs near the transition between the high-diversity and low-diversity regimes and is an example of the well-known phenomenon of “critical slowing down”. We now explicitly acknowledge this critical slowing down in a paragraph in the new section on connections to the chemostat. We also address the question of finite timescales by probing the behavior of our model when nutrients are not completely depleted in each batch. We show in a new figure (Appendix 7—figure 2) that coexistence holds even when substantial portions of the nutrients are left unconsumed (in the figure, coexistence holds until about 45% of the nutrients are unconsumed).

2) Many of the other results besides the two highlighted above could be organized either in the appendix, or as subsections of a final Results section, where the authors examine how relaxing some of the simplifying assumptions made in the model (e.g. the exploration of a softer trade-off imposed by unequal enzyme budgets, the regulation of enzyme expression, the possibility of cross-feeding, the case of unequal K_i_ and Y_i_, or the effect of including the stochastic nature of dilutions) affect their conclusions. If the authors prefer to include it as a Results section in the main text, they should introduce this section by explicitly alerting the reader that its purpose is to check the effect of relaxing the simplifying assumptions they previously made on the main results of the paper, listing them before going into details. That will help the reader understand the connection of the various subsections to the main results.

We have acted on the reviewers’ suggestion. We have created a new Results section where we explicitly alert the readers that the purpose of that section is to relax some of the simplifying assumptions made earlier in the manuscript. We have moved the analyses of unequal K_i_ and Y_i_, cross-feeding, population bottlenecks and unequal enzyme budgets to this new section.

3) Reviewers appreciated the Mutation/Selection section, but they all agreed that it does not add much to the narrative. Cutting it off the paper would help to maintain the focus and streamline the narrative. The authors could develop it further in a separate paper, as it is an interesting and relevant topic

We have followed the reviewers’ suggestion and have cut the mutation/selection section from the manuscript, and intend to publish it separately. We have also cut the “adaptation” section for the same reason, since we consider the “adaptation” section the least relevant for the rest of the manuscript.

4) The comparison to experimental results should be revised. The analysis performed (only one species, not an ecosystem) in Figure 7 does not work as much as a "validation" of the main predictions of the paper, but rather as "support of the model assumptions". The authors described this as: "Qualitatively, the data support the assumptions of substitutable and simultaneous nutrient consumption, while the strong quantitative fit to our model supports the assumption that enzyme strategies do not vary significantly within a batch." The reviewers concurred that validation experiments of the model in this paper would require testing the predictions of remapping: e.g. how co-existing species change as a function of nutrient bolus size. The paper is valid and interesting as it is and without an experimental validation of this prediction, so the current discussion on experiments should be either eliminated or moved to the appendix and changed in tone to reflect its value as a check of some of the model assumptions.

We accept the reviewers’ criticism and have moved the current discussion of experiments to the appendix, changing its tone to reflect its value as a check of some of the model assumptions. As a result, there is a new Appendix 6, which includes all the relevant details of the experimental comparison. This appendix is referred to in the last paragraph of the Discussion.